# A Process Algebraic Approach to Predict and Control Uncertainty in Smart IoT Systems for Smart Cities Based on Permissible Probabilistic Equivalence

**DOI:** 10.3390/s24123881

**Published:** 2024-06-15

**Authors:** Junsup Song, Dimitris Karagiannis, Moonkun Lee

**Affiliations:** 1Department of Computer Science and Engineering, Jeonbuk National University, Jeonju 561-756, Republic of Korea; junsup@jbnu.ac.kr; 2Research Group Knowledge Engineering, University of Vienna, 1010 Vienna, Austria; dimitris.karagiannis@dke.univie.ac.at

**Keywords:** formal method, process algebra, dTP-Calculus, probabilistic equivalences, smart IoT systems, smart city, SAVE, ADOxx

## Abstract

Process algebra is one of the most suitable formal methods to model smart IoT systems for smart cities. Each IoT in the systems can be modeled as a process in algebra. In addition, the nondeterministic behavior of the systems can be predicted by defining probabilities on the *choice* operations in some algebra, such as PALOMA and PACSR. However, there are no practical mechanisms in algebra either to measure or control uncertainty caused by the nondeterministic behavior in terms of satisfiability of the system requirements. In our previous research, to overcome the limitation, a new process algebra called dTP-Calculus was presented to verify probabilistically the safety and security requirements of smart IoT systems: the nondeterministic behavior of the systems was defined and controlled by the static and dynamic probabilities. However, the approach required a strong assumption to handle the unsatisfied probabilistic requirements: enforcing an optimally arbitrary level of high-performance probability from the continuous range of the probability domain. In the paper, the assumption from the previous research is eliminated by defining the levels of probability from the discrete domain based on the notion of *Permissible Process and System Equivalences* so that satisfiability is incrementally enforced by both *Permissible Process Enhancement* in the process level and *Permissible System Enhancement* in the system level. In this way, the unsatisfied probabilistic requirements can be incrementally enforced with better-performing probabilities in the discrete steps until the final decision for satisfiability can be made. The SAVE tool suite has been developed on the ADOxx meta-modeling platform to demonstrate the effectiveness of the approach with a smart EMS (emergency medical service) system example, which is one of the most practical examples for smart cities. SAVE showed that the approach is very applicable to specify, analyze, verify, and especially, predict and control uncertainty or risks caused by the nondeterministic behavior of smart IoT systems. The approach based on dTP-Calculus and SAVE may be considered one of the most suitable formal methods and tools to model smart IoT systems for smart cities.

## 1. Introduction

### 1.1. Smart System

As technology evolves, ICT (information communication technology) systems evolve too. For example, legacy systems evolve toward smart systems. Recently, smart systems have been developed for a set of automated functions with sensors and actuators interconnected by the Internet and supported by various high-technical services, such as AI (artificial intelligence), big data, IoT, cloud computing, etc. [1,2,3,4].

One of the main characteristics of smart systems is the capability to cope with nondeterministic situations from which the systems are subject to make “smart” decisions automatically compared to legacy systems. In that perspective, nondeterminism is a critical issue to investigate the causes of uncertainty and complexity that the systems are facing to provide smart services by means of the Internet generated by the following reasons [5,6,7,8]:Unreliable interactions—smart systems are interacting with other systems or processes throughout the Internet, which are not always predictable, due to internet failure or congestion, causing deadlock.Environmental factors—the environmental changes may influence the operations of smart systems. For example, autonomous driving systems may be influenced by weather or traffic conditions.

Consequently, smart systems may not be always predictable due to the interactions and the conditions caused by nondeterminism. In other words, the systems do not follow fully predefined rules and patterns for every occasions. Therefore, it is quite difficult to predict fully every outcome or behavior of the systems, which may cause critical problems for the reliability and soundness of the smart systems.

In order to handle the nondeterminism, a new methodology for the systems is demanded in a reliable and sound manner to model the nondeterminism at the time of designing the systems and to predict and control the uncertainty caused by the nondeterminism.

### 1.2. Process Algebra in Digital Twins for Smart Cities

One of the main objectives of smart systems is to minimize a variety of risks caused by lack of “smart” capabilities of specific components in the systems. In case of smart IoT systems, which is one of the typical applications of smart systems, it is necessary to define and control the risks caused by the nondeterministic interactions among IoT devices and the unpredictable malfunctions of each device [9,10].

As shown in Figure 1, process algebra is one of the most suitable formal methods to model smart IoT systems since each device can be represented by a process in algebra from the perspective of digital twins in a way that the device in the physical world can perform specific tasks for smart IoT systems as a process in the cyber world.

Digital twins for smart IoT systems can be realized in the following manners:The communication and movements of IoT devices can be represented by τ and δ actions in algebra, respectively [11];The decision and risk can be presented by nondeterministic choice operations with probability in algebra, respectively [11].

As noticeable in the figure, process algebra represents the systems as a set of processes in the cyber world; the decisions of IoT devices from the physical world can be represented by the choice operations in each device.

In addition, in order to predict and control the risks at the systems in the physical world, it is necessary to define the probabilities of the nondeterministic choice operations in the corresponding processes in the cyber world, implying the possibilities of risks, as well as their degrees.

The most ideal systems are the ones without any risk. However, in the case of very complex systems like smart IoT systems, the risks caused by the nondeterministic interactions among their components generated by composed choice operations may be not fully predictable. In this case, it is necessary to control the systems to minimize or eliminate the risks. In other words, it is necessary to control the nondeterminism causing the risks.

### 1.3. Previous Research on Probabilities

The research reported in the paper is based on previous research to propose a new methodology to control the nondeterminism of choice operation for smart systems, as shown in Figure 2 with the white block arrows. In previous research, a new process algebra called dTP-Calculus was presented to verify probabilistically the safety and security requirements of the smart systems, with which the nondeterministic behavior of the systems is defined and controlled by the static and dynamic probabilities. In addition, as a result of the verification, in case of the probabilistic requirements not being satisfied, it was shown, as a feasibility ground, that the probabilities making the decisions were enhanced at some arbitrarily high level so that the requirements were to be satisfied. The detailed steps of the approach in the previous research are as follows [11]; note that each step is indicated with the numbers from the figure:(1)(1-1) The probabilistic OReqs (operational requirements) of smart IoT systems are defined by dTP-Calculus with the ITL (In-The-Large) system model and ITS (In-The-Small) process models. The nondeterminism of the systems is determined by the probabilistic choice operations in the process models.(2)(1-2) A probabilistic EX (execution) model is determined from the ITL model and its ITS models, and all the execution paths are generated. The nondeterministic behavior of the system is constructed by the compositions of the probabilistic choice operations among the ITS models.(3)(1-3) The simulation outcome of each execution path is generated in the form of a GTS (geo-temporal space) model. Each behavior of the system is represented by the composite value of probabilities from the corresponding probabilistic choice operations between the interacting ITS process models.(4)(1-4) The SSReqs (safety and security requirements) are defined by GTS visual logic. Furthermore, the requirements are defined with probabilities too.(5)(1-5) The SSReqs are analyzed and verified on GTSs, especially with respect to the probabilities of the SSReq over the probabilities in the probabilistic EX models.(6)(1-6) In the case where the SSReqs are not satisfied probabilistically, the probabilities of the choice operations in the ITS process models are enhanced to a certain high level, repeating the previous steps until all the SSReqs are satisfied probabilistically.

**Figure 2 sensors-24-03881-f002:**
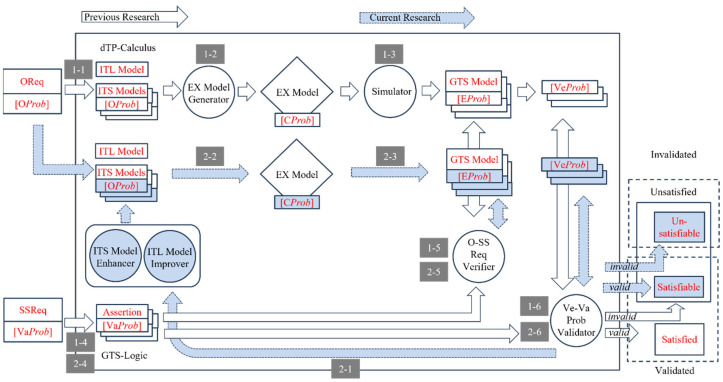
Overview of probabilistic equivalences and incremental improvement methodology.

The approach in the previous research showed the feasibility of the research goal, which is to define and control nondeterminism in the systems with probability so that the probabilistic requirements can be satisfied nondeterministically in the system. However, the approach reveals the following limitations beyond the feasibility:Continuousness—the value of probability was defined to some value in the continuous range of values between zero and one that is conceptually away from the real situation;Arbitrariness—the target value for the new probability to satisfy the requirements was arbitrarily determined without any logical correlation to the physical world.

In order to overcome the limitations, this paper proposes a new methodology for a set of new notions for probabilistic process and system equivalences, from which the nondeterminism in the systems can be enhanced systematically, that is, discretely and incrementally, until all the SSReqs are satisfied probabilistically.

### 1.4. Approach

This paper proposes a more tangible approach, extended from the previous research, to control the nondeterminism in smart IoT systems with both discretely and incrementally controllable probabilities to satisfy the probabilistic SSReqs (safety and security requirements), as shown in Figure 2 with the dark block arrows. The overview of the approach is as follows:(1)*Probabilistic equivalence* is the basic concept used to verify probabilistic requirements as a justification for the enhancement of probability, which overcomes the limitation of the arbitrariness from the previous approach. The equivalences can be classified into two classes.
*Process equivalence* is for the equivalence between two processes representing two IoT devices in smart systems. It implies that two processes are statically identical and are dynamically bisimulative within the range of a ∆ probability difference or increment;*System equivalence* is for the equivalence between two systems representing two smart IoT systems. It implies that two systems are statically identical and are dynamically bisimulative within the range of a set of ∆ probability differences or increments for their corresponding processes.(2)(2-1) Incremental enhancement/improvement is the approach to improve the performance or capability of smart IoT systems to the level where all the SSReqs are satisfied probabilistically. Basically, the system improvement is based on process enhancement.
*Incremental process enhancement* is the enhancement of a process representing each IoT device by increasing the values of probabilities on nondeterministic choice operations, representing replacing the units of the IoT device with a higher version, in terms of the *probabilistic process equivalence*;*Incremental system improvement* is the improvement for a system representing each smart IoT system by replacing its processes with the incrementally enhanced processes in terms of the *probabilistic system equivalence*.

As a result of validation, probabilistic SSReqs are classified into the following categories, as shown on the right side of Figure 2:(1)*Satisfied*—the probability for SSReqs is in the range of validation probability and the validation is completed;(2)*Unsatisfied*—the probability for SSReqs is out of the range of validation probability; it goes through incremental enhancement and improvement, then another round of validation performs on SSReq until the final decision is made.
*Satisfiable*—the new probability for SSReqs is in the range of validation probability and the validation is completed;*Unsatisfiable*—the new probability for SSReqs is out of the range of validation probability.


If probabilistic SSReqs are unsatisfiable at the end, a new system has to be developed for the SSReqs.

### 1.5. Proof of Concepts

In order to prove the feasibility of the approach in the paper, we developed a visual tool suite called SAVE (Specification, Analysis, Verification, Evaluation), and applied the approach to a smart EMS (emergency medical service) system [11,12].

Here, the smart EMS system is the system to improve the quality and efficiency of EMS using IoT technology. It not only increases the survival rate of the critical patients but also provides medical personnel with instantaneous decision-making support.

SAVE is a modeling tool suite to specify, analyze, verify and evaluate a varied set of requirements for DM-RTSs (Distributed Mobile Real-time Systems) with dTP-Calculus and GTS logic, currently focusing on applications in the area of smart IoT systems in smart cities.

SAVE has been developed on the ADOxx meta-modeling platform, with the support of the visualization capability of which all the tools of SAVE are visual, meaning that all the steps of specification, analysis, verification and evaluation are performed visually with the following components [13]:Modeler—It allows the visual specification of the operational requirements of the target smart IoT system with dTP-Calculus, consisting of the ITL (In-The-Large) system and ITS (In-The-Small) process models. The ITL model defines a set of processes in the system, interconnected with a set of communication channels, and the ITS model defines a set of interactions, that is, communication and movements, with other processes in the system. The nondeterministic behavior of the processes is defined by the probability of the choice operation.Generator—It generates an EX (execution) model for the system from a ITL system model with its ITS process models. The EX model contains all the execution paths possible for the system. The probabilistic behavior of the system is defined by generating execution paths with probabilities, composing nondeterministic interactions between processes with probabilities.Simulator—It simulates any selected execution path from the EX model and generates a GTS (geo-temporal space) model. The simulation is performed probabilistically as generated by the Generator for the EX model.Verifier—All the SSReqs (safety and security Requirements) are specified with GTS logic and verified by the Verifier. All the requirements are verified with respect to their corresponding probabilities.Validator—All the SSReqs with probabilities are validated to the system probabilities. If the probabilities are not satisfied, the system improvement starts with process enhancement.

If all the requirements are satisfied, we can predict that the system is reliable and sound. If not, we have to go through a sequence of steps, that is, Steps 2-1 thorough 2-6 in Figure 2, to improve the performance and capabilities of the systems until the requirements are satisfied probabilistically, based on the notions of process and system equivalences, by enhancing their corresponding probabilities to reduce respective faults or risks.

If the requirements are not satisfied at the most enhanced levels of probabilities, the final decision can be made that the system cannot satisfy the requirements probabilistically with any means of upgrading the equivalent processes and their corresponding system.

### 1.6. Contribution

This paper presented a methodology to predict and control the nondeterministic behavior of smart IoT systems with dTP-Calculus based on the notion of process and system equivalences. The methodology is accomplished by the SAVE (Specification, Analysis, Verification, Evaluation) tool suite and the incremental enhancement and improvement approach. The paper demonstrates its feasibility with a smart EMS (emergency medical service) system as an example of smart IoT systems.

The methodology and approach can be considered one of the most efficient practices in the area of process algebra to predict and control uncertainty and risks caused by the nondeterministic behavior of smart IoT systems with probability.

### 1.7. Organization

The paper is organized as follows: Section 2 presents the basic theory for dTP-Calculus and the notions of probabilistic process and system equivalences. Section 3 presents a conceptual approach with the PBC (Producer–Buffer–Consumer) example and incremental enhancement and improvement. Section 4 presents the implementation of the approach with SAVE (Specification, Analysis, Verification, Evaluation) for the smart EMS (emergency medical service) system example. Section 5 presents a comparative study with previous research, including process algebra with respect to equivalences. Finally, conclusions are made with future research topics in Section 6.

## 2. Theoretical Background

### 2.1. dTP-Calculus

dTP-Calculus is a formal method based on process algebra in order to model movements of processes with the following properties for *Distributed Mobile Real-Time Systems* (DM-RTSs):d (delta)—movement;T—time;P—probability.

One of the most suitable applications for dTP-Calculus can be smart IoT systems to represent the interactions and movements of the IoT with respect to mobility, temporality, and probability [11].

#### 2.1.1. Properties of dTP-Calculus

The distinctive characteristics of dTP-Calculus are mobility, temporality, probability, priority, and synchronicity.

##### Mobility Property

In dTP-Calculus, the movements of processes are characterized by the subjectivity and directivity of the movements as follows:Subjectivity—There are two types of movements in dTP-Calculus, (1) active movement, where a process moves autonomously by itself, and (2) passive movement, where a process is moved heteronomously by another process;Directivity—There are two types of movements in dTP-Calculus, (1) inward movement, where a process moves into another sibling process, and (2) outward movement, where a process moves out of its parent process.

With respect to the subjectivity and the directivity of the movements, there are a total of four possible types of movements defined in dTP-Calculus, as shown in Table 1 and listed as follows:*In*—A subjective process moves autonomously into another sibling process and the subjective process needs to obtain permission from the sibling process;*Out*—A subjective process moves autonomously out of its parent sibling process and the subjective process needs to obtain permission from the parent process;*Get*—An objective process is moved into another sibling process heteronomously by the sibling process and the sibling process needs to obtain permission from the subjective process;*Put*—An objective process is moved out of its parent process heteronomously by the parent and the parent process needs to obtain permission from the objective process.

**Table 1 sensors-24-03881-t001:** Movement types in dTP-Calculus.

Movement Types	Subjectivity	Directivity	Visual Presentation
*In*	Autonomous (Active)	Inward	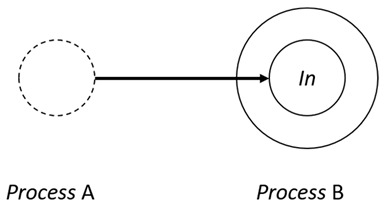
*Out*		Outward	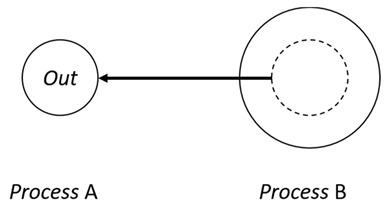
*Get*	Heteronomous (Passive)	Inward	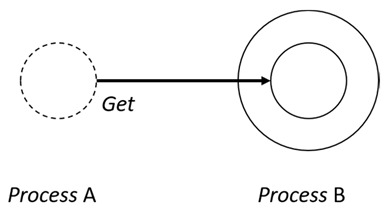
*Put*		Outward	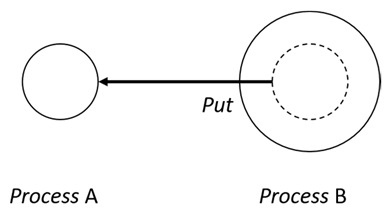

##### Temporality Property

dTP-Calculus defines the following temporal properties in order to specify the temporal requirements for the interactions and movements of processes, which are expressed in discrete and positive integer values, in general as follows:*Ready Time*—The minimum time that a process must wait before performing a specific action, that is, interaction or movement;*Timeout*—The maximum time that a process must wait before performing a specific action, that is, interaction or movement;*Execution Time*—The time that a process consumes to execute a specific action;*Deadline*—The time within which a process must finish a specific action;*Period*—The time that a process repeats a specific action iteratively, like recursion.

##### Probability Property

dTP-Calculus defines the following probability properties for the choice operation, called the probabilistic choice operation, in order to control the nondeterministic choice operation:
Discrete distribution—It allows to specify the choice to be made for its alternatives in the range of discrete values, where the summation of all the alternatives of the choice is 100%;Normal distribution—It allows the choice to be distributed normally for its alternatives with a mean (μ) and a deviation (σ), whose dense function is defined as $\frac{1}{\sigma \sqrt{2\pi }}exp\left(-\frac{{\left(x-\mu \right)}^{2}}{2\sigma^{2}}\right)$;Exponential distribution—It allows the choice to be distributed exponentially for its alternatives with a frequency (λ), whose dense function is ${\lambda e}^{-\lambda x}$;Uniform distribution—It allows the choice to be distributed uniformly for its alternatives in the range of $\left[l,\;u\right]$, whose dense function is $\left\{\begin{array}{c} 0\\ \frac{1}{u-l}\;\left(l\leq x\leq u\right) \end{array}\right.$.


##### Priority Property

Priority in dTP-Calculus is the property assigned to each process utilized to access or control another process, that is, interactions for communication or movements.

It is used as a condition to control asynchronicity between asynchronous interactions or to make a decision on which synchronous action is to be executed among multiple synchronous interactions.

In other words, it is a critical criterion that makes a decision on which action of a process is to be executed first among competing processes.

##### Synchronicity Property

The movements in dTP-Calculus are accomplished synchronously and require synchronous request and permission interactions, as shown in Table 2.

#### 2.1.2. dTP-Calculus Syntax

Table 3 shows the full syntax of dTP-Calculus. Note that P, Q, and E imply a process, respectively, and A indicates an action that a process performs.

It is one of the actions for *A* shown in (1) and (16) of the table: null (empty (16)), communication (send (17), receive (18)), movement ((19): (23)~(26)), control ((20): (27)~(29)).
(a)Empty (16)—It represents a null action, where no action occurs.(b)Communication (send (17), receive (18))—It represents a synchronous communication between sender and receiver. In order to perform a communication between two processes, a set of synchronous actions, that is, send and receive, must occur in an overlapping time between them through a communication channel with the same type of message.(c)Movement (19)—It represents a movement interaction between a requesting process (21) and a permitting process (22). In order to perform a movement, the request action for the movement from the requesting process must be granted by the corresponding permitting action for the movement from the permitting process. There are a total of four types of such actions in dTP-Calculus—*In* (23), *Out* (24), *Get* (25), and *Put* (26).(d)Control (20)—It represents a set of control actions as follows:
(i)*New* (27)—It represents an action for a process to create its child process. The child process cannot have a priority that is higher than that of the parent.(ii)*Kill* (28)—It represents an action for a process to terminate another process. The former must have a priority that is higher than that of the latter.(iii)*Exit* (29)—It represents an action for a process to terminate itself. The child processes with less priorities will be terminated in depth-first order. If there is any child process with a higher priority, this action must be propagated until the child process terminates.

Timed Action (2)—It represents a time action, where the action is specified with the temporal properties [*r*, *to*, *e*, *d*], representing *ready time*, *timeout*, *execution time*, and *deadline*, respectively, and the recursion properties *p* and *n*, representing the period and number of repetitions, respectively.Timed Process (3)—This is the same notion for a process, instead of an action.Priority (4)—It represents the priority that has been stated in the previous subsection. It is expressed with a set of positive integers, where the high number represents a higher priority. Exceptionally, zero represents the highest priority. Note that, as described in the previous subsection, it is used to control asynchronous interactions and multiple synchronous situations.Nesting (5)—It represents inclusion relations among processes, where *P* includes *Q*. The included processes are controlled by an including process. If an inner process has a higher priority than that of its parent, that is, the including process, the inner process can move autonomously out of the including process, that is, its parent. All the processes, that is, parent and child, are running concurrently. When a parent process moves, its child processes move accordingly as included.Channel (6)—It represents a list of channels among processes. A communication occurs as a channel between processes.Choice (7)—It represents a choice operation, a branch of which is nondeterministically taken by a process for an action or interaction. For example, one of *P* and *Q* performs its action nondeterministically.Probabilistic Choice (8)—It represents a probabilistic choice operation, a branch of which is nondeterministically, but probabilistically, taken by a process for an action or interaction. The probabilistic distributions are defined in (12), (13), (14) and (15), respectively, as described in the previous subsection.Parallel (9)—It represents a number of independent processes to be run concurrently at the same time period.Exception (10)—It represent the exceptional handling actions or processes to be performed at the time of an exception being occurred, for example, the violation of temporal requirements for an action or a process.Sequence (11)—It represents the sequential execution of actions in temporal order.

#### 2.1.3. dTP-Calculus Semantics

The semantics of dTP-Calculus are defined by transition rules, as follows:(1)$$Transition=\frac{p_{1},\ldots ,p_{k}}{q}(r)$$

Note that $p_{1},\ldots ,p_{k}$ are premises and *q* is a conclusion under the side condition *r*. It implies that the conclusion can be made from the premises under the condition.

Table 4 shows the transition rules for all the operations defined by the syntax of dTP-Calculus. Brief descriptions of the rules are provided below.
Sequence (1)—Process *P* can perform its action without any premise or condition.ChoiceL, ChoiceR (2)—Only the chosen process can perform its own action as a premise without any condition.Probability Choice (3)—Only the chosen process can perform its own action probabilistically as a premise without any condition under a specified probability distribution.ParlL, ParlR (4)—Each process is independent and runs concurrently without interference as a premise without any condition.Parcom (5)—Synchronous communication occurs between two processes, *P* and *Q*. As a result of the communication interaction, each process makes its respective transition, as shown in the premises, without any side condition.NestingO, NestingI (6)—Both nesting and nested processes can perform their own transitions independently, as defined in the premises without any side condition.NestingCom (7)—A synchronous communication interaction is possible between the nesting and nested processes.In (8), Out (9), Get (10), Put (11)—Each movement is defined as a *synchronous* interaction between a requesting process and a permitting process for the action. As stated, *In* and *Out* are for autonomous movements and *Get* and *Put* are for heteronomous movements. Note that the inclusion relations are modified as a result of the movements.InP (12), OutP (13), GetP (14), PutP (15)—Each movement is defined as an *asynchronous* interaction between a requesting process and a permitting process for the action. The difference is that if the request process has a higher priority, it does not have to wait for permission from the permitting process, but it can move asynchronously without permission.TickTimeR (16)—As time ticks with ${⊳}_{t}$, the ready time *r* and deadline *d* are elapsed by the ticks ${⊳}_{t}$.TickTimeTO (17)—As time ticks idly with ${⊳}_{t}$, timeout *to* and deadline *d* are elapsed by the ticks ${⊳}_{t}$.TickTimeEnd (18)—An action is completed after the execution time *e*.TickTimeSyncE (19)—As time ticks for the synchronous interaction with ${⊳}_{t}$, the execution time *e* and deadline *d* are elapsed by the ticks ${⊳}_{t}$.TickTimeAsyncE (20)—As time ticks for the asynchronous interaction with ${⊳}_{t}$, the execution time *e* and deadline *d* are elapsed by the ticks ${⊳}_{t}$ after ready time *r*.Timeout (21)—When timeout *to* becomes zero, a fault occurs. Exception handler *P* should be activated.Deadline (22)—When deadline *d* becomes zero, a fault occurs. Exception handler *P* should be activated.Period (23)—After recursion, repetition *n* is decremented by one.PeriodEnd (24)—If repetition *n* becomes zero, the recursion ends.

#### 2.1.4. dTP-Calculus Rules

dTP-Calculus rules are defined in Table 5 and described as follows:Choice1 (1), Choice2 (2), Choice3 (3)—commutative and distributive rules for the choice operation;Parallel1 (4), Parallel2 (5), Parallel3 (6)—commutative and associative rules for parallel operation;Nesting1 (7), Nesting2 (8)—associative rules for the choice operation between processes nested in another process.Distributive1 (9), Distributive2 (10)—distributive rules for parallel over choice are only available in sibling relations. If not, a sequencing problem occurs.

**Table 5 sensors-24-03881-t005:** dTP-Calculus rules.

Type	Rules	Index
Choice1	P+P=P	(1)
Choice2	P+Q=Q+P	(2)
Choice3	$\left(P+Q\right)+R=P+(Q+R)$	(3)
Parallel1	P|ϕ=P	(4)
Parallel2	P|Q=Q|P	(5)
Parallel3	$\left(P|Q\right)|R=P|(Q|R)$	(6)
Nesting1	P[ϕ]=P	(7)
Nesting2	$R\left[P\right]+R[Q]=R[P+Q]$	(8)
Distributive1	$P|\left(Q+R\right)=\left(P|Q\right)+(P|R)$	(9)
Distributive2	$\left(a_{1}+a_{2}\right).P=a_{1}.P+a_{2}.P$	(10)

### 2.2. Probabilistic Equivalences

As stated in the Introduction, there are two types of equivalences for processes and systems.

#### 2.2.1. Probabilistic Process Equivalences

Process P is defined as P=(S,T), where S, T are two sets of states and transitions between two states, represented by $S=\left\{s_{1},\cdots ,s_{f_{s}}\right\}$ and $T=\left\{t_{1},\cdots ,t_{f_{t}}\right\}$, where each $t=\langle s_{i},s_{j}\rangle$ and $s_{i},s_{j}\in S$. For example, Process R is defined as follows in dTP-Calculus. *R* is defined as two sets of states and transitions between states. Note that the diagram for the process is shown for this example only since the diagram is obvious and will be shown in the ITS (In-The-Small) model of the SAVE (Specification, Analysis, Verification, Evaluation) tool in the following implementation section.



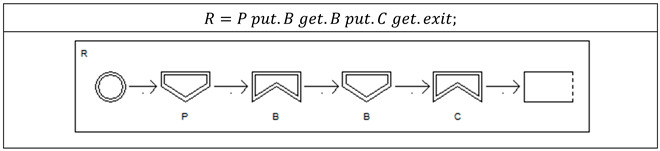



If two processes $P_{i}$ and $P_{j}$ are identical, that is, $P_{i}=(S_{i},T_{i})$, $P_{j}=(S_{j},T_{j})$, $S_{i}=S_{j}$, and $T_{i}=T_{j}$, then two processes are *equivalent*, that is, $P_{i}=P_{j}$. For example, Processes $R_{1}$ and $R_{2}$ are defined as follows in dTP-Calculus and the two processes are identical, that is, $R_{1}=R_{2}$:R1=P put.B get.B put.C get.exit;R2=P put.B get.B put.C get.exit.

Process P is defined to be *probabilistic*, if Process $P=(S,T^{p})$, where S and $T^{p}$ are two sets of states and transitions between two states, represented by $S=\left\{s_{1},\cdots ,s_{f_{s}}\right\}$ and $T^{p}=\left\{t_{1}^{p},\cdots ,t_{f_{t}}^{p}\right\}$, where, for each $t_{i}^{p}={\langle s_{j},s_{k}\rangle }^{p}$ and $s_{i},s_{j}\in S$, p is the probability of the successful transition between $s_{i}$ and $s_{j}$. For example, Process P is defined as follows in dTP-Calculus and *R* is defined as two sets of states and transitions between states, especially a choice operation with probability:P=PBSend R1¯0.5.put R1.put R2+PBDSend R2¯0.5.put R2.put R1.exit;

If two probabilistic processes $P_{i}$ and $P_{j}$ are probabilistically identical, that is, $P_{i}=(S_{i},T_{i}^{p})$, $P_{j}=(S_{j},T_{j}^{p})$, $S_{i}=S_{j}$, and $T_{i}^{p}=T_{j}^{p}$, then the two processes are *probabilistically equivalent*, that is, $P_{i}\sim P_{j}$. For example, Processes $P_{1}$ and $P_{2}$ are defined as follows in dTP-Calculus and the two processes are probabilistically identical, that is, $P_{1}∼P_{2}$, especially with respect to the choice operations with the same probabilities:P1=PBSend R1¯0.5.put R1.put R2+PBD1Send R2¯0.5.put R2.put R1.exit;P2=PBSend R1¯{0.5}.put R1.put R2+PBD2Send R2¯{0.5}.put R2.put R1.exit.

If the difference between two probabilities $p_{i}$ and $p_{j}$ are within $\Delta^{{+}_{D}}$, then the two processes are within the *permissible probability* range, that is, $P_{i}\Delta^{{+}_{D}}P_{j}$. For example, Processes $P_{1}$ and $P_{2}$ are identical, but not probabilistic, and the permissible range for $P_{1}$ and $P_{2}$ is defined as follows: $P_{1}\left(\left[0.5{+}_{D_{1}}0.5\right]\right)\Delta^{<0.4>}P_{2}\left(\left[0.9{+}_{D_{2}}0.1\right]\right)$.
P1=PBSend R1¯0.5.put R1.put R2+PBD1Send R2¯0.5.put R2.put R1.exit;P2=PBSend R1¯{0.9}.put R1.put R2+PBD2Send R2¯{0.1}.put R2.put R1.exit.

The difference between the probabilities in their choice operations is defined as the difference from the former to the latter. $\Delta^{{+}_{D}}$ can be extended to the set of the differences among the corresponding probabilities between two probabilistic processes.

If two processes $P_{i}=(S_{i},T_{i}^{p})$ and $P_{j}=(S_{j},T_{j}^{p})$ are probabilistic, all the transitions of $P_{i}$ and $P_{j}$, that is, $T_{i}^{p}=\left\{t_{i,1}^{p_{i,1}},\cdots ,t_{i,n}^{p_{i,n}}\right\}$ and $T_{j}^{p}=\left\{t_{j,1}^{p_{j,1}},\cdots ,t_{j,n}^{p_{j,n}}\right\}$, and their corresponding probabilities for $t_{i,k}^{p_{i,k}}$ and $t_{j,k}^{p_{j,k}}$, where 1≤k≤n, that is, $p_{i,k}\Delta^{p}p_{j,k}$, where 1≤k≤n, then two processes are *permissibly-probabilistically equivalent*, that is, $P_{i}\overset{\Delta^{p}}{\sim }P_{j}$. For example, Processes $P_{1}$ and $P_{2}$ are identical, but not probabilistic, as shown below. If the permissible probability $\Delta^{{+}_{D}}$ is defined to be $\Delta^{<0.2>}$, then Processes $P_{1}$ and $P_{2}$ are permissibly probabilistic–equivalent in $P_{1}\left(\left[0.5{+}_{D_{1}}0.5\right]\right)\overset{\Delta^{<0.2>}}{∼}P_{2}\left(\left[0.7{+}_{D_{2}}0.3\right]\right)$.
P1=PBSend R1¯0.5.put R1.put R2+PBD1Send R2¯0.5.put R2.put R1.exit;P2=PBSend R1¯{0.7}.put R1.put R2+PBD2Send R2¯{0.3}.put R2.put R1.exit.

#### 2.2.2. Probabilistic System Equivalences

System S is defined to be a set of processes, that is, $S=\left\{P_{k}|1\leq k\leq n\right\}$. For example, System $P_{1}B_{1}C_{1}$ is defined as follows in dTP-Calculus, and its system view is shown as well:$$\begin{array}{l} P_{1}B_{1}C_{1}=P_{1}[R1∥R2]∥B_{1}∥C_{1}\\ P_{1}=\left(PB\left(\bar{Send\;R1}\right).put\;R1.put\;R2+\;PB\left(\bar{Send\;R2}\right).put\;R2.put\;R1\right).exit;\\ B_{1}=\left(PB\left(Send\;R1\right).get\;R1.get\;R2+\;PB\left(Send\;R2\right).get\;R2.get\;R2\right).\\ \;\left(CB\left(Send\;R1\right).put\;R1.put\;R2+\;CB\left(Send\;R2\right).put\;R2.put\;R1\right).exit;\\ C_{1}=\left(CB\left(\bar{Send\;R1}\right).get\;R1.get\;R2+\;CB\left(\bar{Send\;R2}\right).get\;R2.get\;R1\right).exit;\\ R1=P_{1}\;put.B_{1}\;get.B_{1}put.C_{1}\;get.exit;\\ R2=P_{1}\;put.B_{1}\;get.B_{1}\;put.C_{1}\;get.exit. \end{array}$$

If two systems $S_{i}=\left\{P_{i,k}|1\leq k\leq n\right\}$ and $S_{j}=\left\{P_{j,k}|1\leq k\leq n\right\}$ are identical, that is, $P_{i,k}=P_{j,k}$ for 1≤k≤n, then two systems are *equivalent*, that is, $S_{i}=S_{j}$. For example, System $P_{2}B_{2}C_{2}$ is defined as follows, and it is identical to $P_{1}B_{1}C_{1}$ in the above example: $P_{1}B_{1}C_{1}=P_{2}B_{2}C_{2}$, since $P_{1}=P_{2}$, $B_{1}=B_{2}$, and $C_{1}=C_{2}$.
$$\begin{array}{l} P_{2}B_{2}C_{2}=P_{2}[R1∥R2]∥B_{2}∥C_{2}\\ P_{2}=\left(PB\left(\bar{Send\;R1}\right).put\;R1.put\;R2+PB\left(\bar{Send\;R2}\right).put\;R2.put\;R1\right).exit;\\ B_{2}=\left(PB\left(Send\;R1\right).get\;R1.get\;R2+PB\left(Send\;R2\right).get\;R2.get\;R2\right).\\ \;\left(CB\left(Send\;R1\right).put\;R1.put\;R2+CB\left(Send\;R2\right).put\;R2.put\;R1\right).exit;\\ C_{2}=\left(CB\left(\bar{Send\;R1}\right).get\;R1.get\;R2+CB\left(\bar{Send\;R2}\right).get\;R2.get\;R1\right).exit;\\ R1=P_{2}\;put.B_{2}\;get.B_{2}\;put.C_{2}\;get.exit;\\ R2=P_{2}\;put.B_{2}\;get.B_{2}\;put.C_{2}\;get.exit. \end{array}$$

System S is defined to be *probabilistic* if $S=\left\{P_{k}|1\leq k\leq n\right\}$ and at least one of the processes $P_{k}$ for 1≤k≤n is probabilistic. For example, System $P_{1}^{\prime }B_{1}^{\prime }C_{1}^{\prime }$ is defined below as a probabilistic system since Processes $P_{1}^{\prime }$, $B_{1}^{\prime }$ and $C_{1}^{\prime }$ are defined as probabilistic processes.
$$\begin{array}{l} P_{1}^{\prime }B_{1}^{\prime }C_{1}^{\prime }=P_{1}^{\prime }[R1∥R2]∥B_{1}^{\prime }∥C_{1}^{\prime }\\ P_{1}^{\prime }=\left(PB\left(\bar{Send\;R1}\right)\left\{0.6\right\}.put\;R1.put\;R2\;{+}_{D}\;PB\left(\bar{Send\;R2}\right)\{0.4\}.put\;R2.put\;R1\right).exit;\\ B_{1}^{\prime }=\left(PB\left(Send\;R1\right)\left\{0.7\right\}.get\;R1.get\;R2\;{+}_{D}\;PB\left(Send\;R2\right)\{0.3\}.get\;R2.get\;R2\right).\\ \;\left(CB\left(Send\;R1\right)\left\{0.5\right\}.put\;R1.put\;R2\;{+}_{D}\;CB\left(Send\;R2\right)\{0.5\}.put\;R2.put\;R1\right).exit;\\ C_{1}^{\prime }=\left(CB\left(\bar{Send\;R1}\right)\left\{0.8\right\}.get\;R1.get\;R2\;{+}_{D}\;CB\left(\bar{Send\;R2}\right)\{0.2\}.get\;R2.get\;R1\right).exit;\\ R1=P_{1}^{\prime }\;put.B_{1}^{\prime }\;get.B_{1}^{\prime }\;put.C_{1}^{\prime }\;get.exit;\\ R2=P_{1}^{\prime }\;put.B_{1}^{\prime }\;get.B_{1}^{\prime }\;put.C_{1}^{\prime }\;get.exit. \end{array}$$

If two probabilistic systems $S_{i}$ and $S_{j}$ are probabilistically identical, that is, for $S_{i}=\left\{P_{i,k}|1\leq k\leq n\right\}$ and $S_{j}=\left\{P_{j,k}|1\leq k\leq n\right\}$, for all the corresponding processes, $P_{i,k}\sim P_{j,k}$ for 1≤k≤n, then the two systems are *probabilistically equivalent*, that is, $S_{i}\approx S_{j}$. For example, System $P_{2}^{\prime }B_{2}^{\prime }C_{2}^{\prime }$ defined below is probabilistically equivalent to the previous system $P_{1}^{\prime }B_{1}^{\prime }C_{1}^{\prime }$ since $P_{1}^{\prime }\sim P_{2}^{\prime }$, $B_{1}^{\prime }\sim B_{2}^{\prime }$, and $C_{1}^{\prime }\sim C_{2}^{\prime }$: $P_{1}^{\prime }B_{1}^{\prime }C_{1}^{\prime }\approx P_{2}^{\prime }B_{2}^{\prime }C_{2}^{\prime }$.
$$\begin{array}{l} P_{2}^{\prime }B_{2}^{\prime }C_{2}^{\prime }=P_{2}^{\prime }[R1∥R2]∥B_{2}^{\prime }∥C_{2}^{\prime }\\ P_{2}^{\prime }=\left(PB\left(\bar{Send\;R1}\right)\{0.6\}.put\;R1.put\;R2{+}_{D_{1}}\;PB\left(\bar{Send\;R2}\right)\{0.4\}.put\;R2.put\;R1\right).exit;\\ B_{2}^{\prime }=\left(PB\left(Send\;R1\right)\{0.7\}.get\;R1.get\;R2{+}_{D_{2}}\;PB\left(Send\;R2\right)\{0.3\}.get\;R2.get\;R2\right).\\ \;\left(CB\left(Send\;R1\right)\{0.5\}.put\;R1.put\;R2{+}_{D_{3}}\;CB\left(Send\;R2\right)\{0.5\}.put\;R2.put\;R1\right).exit;\\ C_{2}^{\prime }=\left(CB\left(\bar{Send\;R1}\right)\{0.8\}.get\;R1.get\;R2{+}_{D_{4}}\;CB\left(\bar{Send\;R2}\right)\{0.2\}.get\;R2.get\;R1\right).exit;\\ R1=P_{2}^{\prime }\;put.B_{2}^{\prime }\;get.B_{2}^{\prime }\;put.C_{2}^{\prime }\;get.exit;\\ R2=P_{2}^{\prime }\;put.B_{2}^{\prime }\;get.B_{2}^{\prime }\;put.C_{2}^{\prime }\;get.exit. \end{array}$$

If the difference between two probabilities $p_{i}$ and $p_{j}$ is within $\Delta^{{+}_{D}}$, then the two systems are within the *permissible probability* range, that is, $S_{i}\Delta^{p}S_{j}$. For example, since the two systems $P_{1}^{\prime }B_{1}^{\prime }C_{1}^{\prime }$ and $P_{2}^{\prime \prime }B_{2}^{\prime \prime }C_{2}^{\prime \prime }$ are identical but not probabilistic, as shown as below, their probabilistic permissibility will be within $D=\langle P_{1}^{\prime }\Delta^{{+}_{D_{P}}}P_{2}^{\prime \prime }\rangle$, that is, $P_{1}^{\prime }B_{1}^{\prime }C_{1}^{\prime }\Delta^{{+}_{D_{P}}}P_{2}^{\prime \prime }B_{2}^{\prime \prime }C_{2}^{\prime \prime }$, where $D_{p}=\langle P_{1}^{\prime }\left(\left[0.6{+}_{D_{1}}0.4\right]\right)\Delta^{<0.1>}P_{2}^{\prime \prime }\left(\left[0.7{+}_{D_{2}}0.3\right]\right)\rangle$, since $B_{1}^{\prime }$, $C_{1}^{\prime }$ are probabilistically equivalent with $B_{2}^{\prime \prime }$, $C_{2}^{\prime \prime }$, respectively, that is, $B_{1}^{\prime }∼B_{2}^{\prime \prime }$ and $C_{1}^{\prime }∼C_{2}^{\prime \prime }$.
$$\begin{array}{l} P_{2}^{\prime \prime }B_{2}^{\prime \prime }C_{2}^{\prime \prime }=P_{2}^{\prime \prime }[R1∥R2]∥B_{2}^{\prime \prime }∥C_{2}^{\prime \prime }\\ P_{2}^{\prime \prime }=\left(PB\left(\bar{Send\;R1}\right)\{0.7\}.put\;R1.put\;R2{+}_{D_{1}}\;PB\left(\bar{Send\;R2}\right)\{0.3\}.put\;R2.put\;R1\right).exit;\\ B_{2}^{\prime \prime }=\left(PB\left(Send\;R1\right)\{0.7\}.get\;R1.get\;R2{+}_{D_{2}}\;PB\left(Send\;R2\right)\{0.3\}.get\;R2.get\;R2\right).\\ \;\left(CB\left(Send\;R1\right)\{0.5\}.put\;R1.put\;R2{+}_{D_{3}}\;CB\left(Send\;R2\right)\{0.5\}.put\;R2.put\;R1\right).exit;\\ C_{2}^{\prime \prime }=\left(CB\left(\bar{Send\;R1}\right)\{0.8\}.get\;R1.get\;R2{+}_{D_{4}}\;CB\left(\bar{Send\;R2}\right)\{0.2\}.get\;R2.get\;R1\right).exit;\\ R1=P_{2}^{\prime \prime }\;put.B_{2}^{\prime \prime }\;get.B_{2}^{\prime \prime }\;put.C_{2}^{\prime \prime }\;get.exit;\\ R2=P_{2}^{\prime \prime }\;put.B_{2}^{\prime \prime }\;get.B_{2}^{\prime \prime }\;put.C_{2}^{\prime \prime }\;get.exit. \end{array}$$

If, for two systems, $S_{i}=\left\{P_{i,k}|1\leq k\leq n\right\}$ and $S_{j}=\left\{P_{j,k}|1\leq k\leq n\right\}$, all the corresponding processes, $\left(P_{i,k}\overset{\Delta^{p}}{\sim }P_{j,k}\right)$ for 1≤k≤n, are permissibly–probabilistically equivalent, then the two systems are *permissibly–probabilistically equivalent*, that is, $\left(S_{i}\overset{\Delta^{p}}{\approx }S_{j}\right)$. For example, the above systems $P_{1}^{\prime }B_{1}^{\prime }C_{1}^{\prime }$ and $P_{2}^{\prime \prime }B_{2}^{\prime \prime }C_{2}^{\prime \prime }$ are identical but not probabilistic. If the permissible probability $\Delta^{{+}_{D}}$ is defined to be $\Delta^{<0.4>}$, where $P_{1}^{\prime }B_{1}^{\prime }C_{1}^{\prime }\Delta^{{+}_{D_{P}}}P_{2}^{\prime \prime }B_{2}^{\prime \prime }C_{2}^{\prime \prime }$ and $D=\langle P_{1}^{\prime }\Delta^{{+}_{D_{P}}}P_{2}^{\prime \prime }\rangle$ with $D_{p}=\langle P_{1}^{\prime }\left(\left[0.6{+}_{D_{1}}0.4\right]\right)\Delta^{<0.1>}P_{2}^{\prime \prime }\left(\left[0.7{+}_{D_{2}}0.3\right]\right)\rangle$, then $P_{1}^{\prime }B_{1}^{\prime }C_{1}^{\prime }\overset{\Delta^{D_{P}\leq 0.4}}{\approx }P_{2}^{\prime \prime }B_{2}^{\prime \prime }C_{2}^{\prime \prime }$.

## 3. Approach

### 3.1. Probabilistic Verification

The execution model for System S is defined as a labelled transition system $E_{S}=\left(S_{S},T_{S}\right)$, where $S_{S},T_{S}$ are two sets of system states, represented by the set of processes states in the system and transitions between the system states, respectively. The transition from a system state to another state occurs as defined by the semantics of dTP-Calculus.

The probabilistic execution model for System S is defined as a labelled transition system $E_{S}^{p}=\left(S_{S},T_{S}^{p}\right)$, where $S_{S},T_{S}^{p}$ are two sets of system states represented by the set of processes states in the system and probabilistic transitions between the system states, respectively. The transition from a system state to another state occurs as defined by the semantics of dTP-Calculus for probabilistic choice operation. For example, Figure 3 shows the probabilistic execution model for the example $P_{1}^{\prime }B_{1}^{\prime }C_{1}^{\prime }$ system. Note that all the transition from one system state to another is labelled with the types of transitions with probability, calculated by the composition of the corresponding synchronous interactions for communication and movements. There are four possible deadlock situations: two between $P_{1}^{\prime }$ and $B_{1}^{\prime }$, and two between $B_{1}^{\prime }$ and $C_{1}^{\prime }$. Since they occur in sequence, there are a total of eight deadlocks possible for the system at the time of execution.

The *Normal Termination Probability* (*ntp*) of a system during execution is obtained by summing the probabilities of all the normal execution paths from the execution model of the system. The *Abnormal Termination Probability* (*atp*) of a system during execution is obtained by summing the probabilities of all the abnormal execution paths from the execution model of the system, such as deadlock cases. For example, in the above example, the *ntp* and *atp* of System $P_{1}^{\prime }B_{1}^{\prime }C_{1}^{\prime }$ are 0.27 and 0.73, respectively.

The safety and security requirements (*SSReqs*) $R^{ssr}$ for System S are defined as a set of classified safety and security requirements for the system during execution. For example, there are two sets of SSReqs $R_{1}^{ssr}$ and $R_{2}^{ssr}$ defined for System $P_{1}^{\prime }B_{1}^{\prime }C_{1}^{\prime }$ in Table 6. Informally, $R_{1}^{ssr}$ and $R_{2}^{ssr}$ are classified as sets of safety and security requirements for the system, respectively.

### 3.2. Probabilistic Validation

The verification of all the SSReqs (safety and security requirements) of a system is performed by the SSReq verifier for all probabilistic execution paths of the probabilistic execution model. The total probability of satisfying the requirements is obtained by summing all the probabilities of the paths where the requirements are satisfied. For example, Table 7 shows the results of the probabilistic verification of all the SSReqs of System $P_{1}^{\prime }B_{1}^{\prime }C_{1}^{\prime }$ listed in Table 6. The satisfaction probabilities for the requirements are 0.18, 0.27, 0.18 and 0.09, respectively, and were obtained as follows. Note that *P_r_* and *P_e_* represent the probabilities for requirements and execution, respectively.

$p_{r}\left(R_{1}^{sc}\right)=p_{e}\left(ePath_{1}\right)+p_{e}\left(ePath_{10}\right)=0.168+0.012=0.18$;$p_{r}\left(R_{2}^{sc}\right)=p_{e}\left(ePath_{1}\right)+p_{e}\left(ePath_{4}\right)+p_{e}\left(ePath_{7}\right)+p_{e}\left(ePath_{10}\right)=0.168+0.168+0.048+0.012=0.27$;$p_{r}\left(R_{1}^{sf}\right)=p_{e}\left(ePath_{1}\right)+p_{e}\left(ePath_{10}\right)=0.168+0.012=0.18$;$p_{r}\left(R_{2}^{sc}\right)=p_{e}\left(ePath_{4}\right)+p_{e}\left(ePath_{7}\right)=0.168+0.048=0.09$.

**Table 7 sensors-24-03881-t007:** Analysis of the probabilistic verification of the requirements of $P_{1}^{\prime }B_{1}^{\prime }C_{1}^{\prime }$.

	*eP*1	*eP*2	*eP*3	*eP*4	*eP*5	*eP*6	*eP*7	*eP*8	*eP*9	*eP*10	Total
τ_1_•τ_2_	τ_1,1_•τ_2,1_	τ_1,1_•τ_2,⸣1_	τ_1,1_•τ_2,⸣2_	τ_1,1_•τ_2,2_	τ_1, ⸣1_	τ_1, ⸣2_	τ_1,2_•τ_2,1_	τ_1,2_•τ_2,⸣1_	τ_1,2_•τ_2,⸣2_	τ_1,2_•τ_2,2_	
Prob.	0.168	0.168	0.042	0.042	0.28	0.28	0.048	0.048	0.012	0.012	1.00
$R_{1}^{sc}$	◯	✕	✕	✕	✕	✕	✕	✕	✕	◯	0.18
$R_{2}^{sc}$	◯	✕	✕	◯	✕	✕	◯	✕	✕	◯	0.27
$R_{1}^{sf}$	◯	✕	✕	✕	✕	✕	✕	✕	✕	◯	0.18
$R_{2}^{sf}$	✕	✕	✕	◯	✕	✕	◯	✕	✕	✕	0.09

Note that $R_{1}^{sc}$ is a subset of $R_{2}^{sc}$, and $R_{2}^{sf}$ and $R_{2}^{sf}$ are disjointed, meaning that there is no dependency among the requirements. However, it is necessary to analyze the dependencies among the requirements in order to prevent some contradictions and discrepancies from being specified during the verification process.

The validation requirement for a safety and security requirement is defined by the probability that the requirement is satisfied in the probabilistic execution model of the target system. For example, the left side of Table 8 shows the validation requirements for all the safety and security requirements of System $P_{1}^{\prime }B_{1}^{\prime }C_{1}^{\prime }$: 0.10, 0.15, 0.10, and 0.05 for $R_{1}^{sc}$, $R_{2}^{sc}$, $R_{1}^{sf}$, and $R_{2}^{sf}$, respectively.

The validation of each probabilistic safety and security requirement is performed by comparing the probability for the validation requirement with the probability for the satisfaction of the requirement in the probabilistic execution model of the target system. If the former is less than the latter, the validation requirement is *satisfied*. If not, it is *unsatisfied*. For example, the validation requirements for all the safety and security requirements of System $P_{1}^{\prime }B_{1}^{\prime }C_{1}^{\prime }$ are satisfied, as shown in Table 8. However, if the validation requirements are tightened, as shown in the left side of Table 9, some requirements become *unsatisfied* for their validation: $R_{1}^{sc}$ and $R_{1}^{sf}$.

### 3.3. Incremental Enhancement and Improvement

In case that there is any probabilistic SSReq (safety and security requirement) that is not valid in the previous step, it is possible to enhance the satisfiability of the probabilistic SSReq by strengthening the operational power of some problematic process by upgrading its version. Once the upgrade is performed, it goes through the validation procedure for the probabilistic SSReq in the newly generated execution model of the system with the upgraded version of the processes. If the validation requirement is satisfied, it is classified as a *satisfiable* probabilistic SSReq for the system with the upgraded processes. If not, meaning that there are no other better versions of the processes that satisfy the probability requirement, the SSReq is classified as an *unsatisfiable* probabilistic SSReq.

*Process enhancement* is defined as increasing the value of probabilities on the transitions in a process defined by the discrete level representing the capability of IoT device in smart systems. Furthermore, *system improvement* is defined as replacing the problematic processes with the enhanced processes to increase the performance or capabilities of the target IoT systems.

For example, as shown in the previous example, there are two SSReqs that are unsatisfied for the existing $P_{1}^{\prime }B_{1}^{\prime }C_{1}^{\prime }$: $R_{1}^{sc}$ and $R_{1}^{sf}$. For enhancement, the problematic process $P_{1}^{\langle 0.60\rangle }$ is to be upgraded to its upper version $P_{2}^{\langle 0.70\rangle }$ by increasing the probability for its choice operation, as shown in Figure 4. The improved system with the new enhanced version of Process $P_{2}^{\langle 0.70\rangle }$ becomes System $P_{2}^{\prime }B_{1}^{\prime }C_{1}^{\prime }$. Figure 5 shows the probabilistic execution model for the improved system, and the corresponding probabilistic SSReqs are shown in Table 10. Now, the satisfaction probabilities for the SSReqs are 0.205, 0.290, 0.205, and 0.085, respectively, and were obtained as follows:
$p_{r}\left(R_{1}^{sc}\right)=p_{e}\left(ePath_{1}\right)+p_{e}\left(ePath_{10}\right)=0.196+0.009=0.205$;$p_{r}\left(R_{2}^{sc}\right)=p_{e}\left(ePath_{1}\right)+p_{e}\left(ePath_{4}\right)+p_{e}\left(ePath_{7}\right)+p_{e}\left(ePath_{10}\right)=0.196+0.049+0.036+0.009=0.290$;$p_{r}\left(R_{1}^{sf}\right)=p_{e}\left(ePath_{1}\right)+p_{e}\left(ePath_{10}\right)=0.205+0.009=0.205$;$p_{r}\left(R_{2}^{sc}\right)=p_{e}\left(ePath_{4}\right)+p_{e}\left(ePath_{7}\right)=0.049+0.036=0.085$.

**Figure 4 sensors-24-03881-f004:**
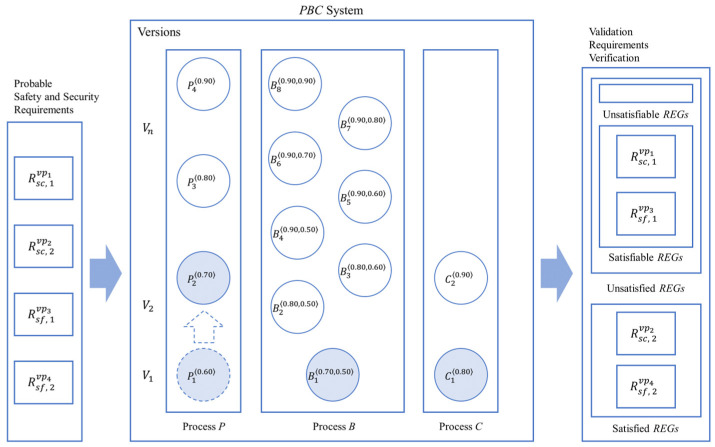
Overview of the incremental system improvement for the PBC (Producer–Buffer–Consumer) example.

**Figure 5 sensors-24-03881-f005:**
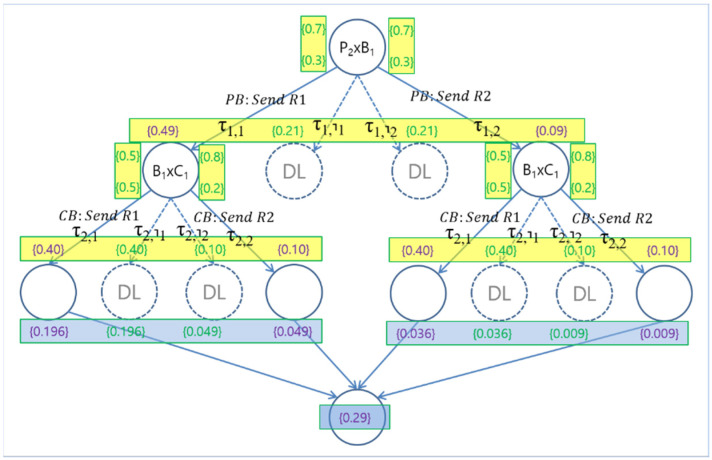
Improved conceptual probabilistic EX (execution) model.

**Table 10 sensors-24-03881-t010:** Analysis of the probabilistic verification of the Requirements of $P_{2}^{\prime }B_{1}^{\prime }C_{1}^{\prime }$.

	*eP*1	*eP*2	*eP*3	*eP*4	*eP*5	*eP*6	*eP*7	*eP*8	*eP*9	*eP*10	Total
τ_1_•τ_2_	τ_1,1_•τ_2,1_	τ_1,1_•τ_2,⸣1_	τ_1,1_•τ_2,⸣2_	τ_1,1_•τ_2,2_	τ_1, ⸣1_	τ_1, ⸣2_	τ_1,2_•τ_2,1_	τ_1,2_•τ_2,⸣1_	τ_1,2_•τ_2,⸣2_	τ_1,2_•τ_2,2_	
Prob.	0.196	0.196	0.049	0.049	0.21	0.21	0.036	0.036	0.009	0.009	1.00
$R_{1}^{sc}$	◯	✕	✕	✕	✕	✕	✕	✕	✕	◯	0.205
$R_{2}^{sc}$	◯	✕	✕	◯	✕	✕	◯	✕	✕	◯	0.290
$R_{1}^{sf}$	◯	✕	✕	✕	✕	✕	✕	✕	✕	◯	0.205
$R_{2}^{sf}$	✕	✕	✕	◯	✕	✕	◯	✕	✕	✕	0.085

Now, as shown in Table 11 for Case 2, the *unsatisfied* SSReqs for System $P_{1}^{\prime }B_{1}^{\prime }C_{1}^{\prime }$, that is, $R_{1}^{sc}$ and $R_{1}^{sf}$, becomes *satisfiable* for System $P_{2}^{\prime }B_{1}^{\prime }C_{1}^{\prime }$.

## 4. Implementation

### 4.1. ADOxx Meta-Modeling Platform

ADOxx is an integrated meta-modeling platform for business & IT (information technology) architecture, process modeling, data modeling, etc. This platform provides a flexible methodology to define modeling languages for various domains. The main features of the environment are [14] as follows:Meta-modeling languages—ADOxx provides a function to develop modeling languages for business domains. It is very useful to utilize the existing standard modeling languages or to develop completely new modeling languages;Meta-modeling modules—ADOxx provides a set of meta-modeling functions to define modeling languages and extend them, thorough which users can define modeling processes more effectively by defining the structures and rules of the modeling languages;Modeling tools—ADOxx provides a set of intuitive tools with which users can design and edit graphically the models and through which the users can understand more visually modeled processes and manage them more effectively.Plugin extension—It allows an extension of functions by supporting various plugin facilities through which the users can extend specific functions, if needed, and integrate the platform with other external systems.

ADOxx is used in various application areas, including business process modeling, enterprise architecture management, information systems development, data modeling, etc. [13,15].

### 4.2. SAVE

SAVE (Specification, Analysis, Verification, Evaluation) is a visual tool suite for dTP-Calculus developed on the ADOxx meta-modeling platform. It is designed originally to specify, analyze, verify and evaluate Distributed Mobile Real-Time Systems (DM-RTSs) with dTP-Calculus, consisting of a modeler, simulator, analyzer, verifier, etc., as shown in Figure 6. Currently SAVE has been applied to the area of smart IoT systems for smart cities [11,12].

#### 4.2.1. SAVE: Modeler

*Modeler* is one of the main tools in SAVE to model processes for the target system, i.e., the IoT in smart IoT systems, generating ITL (In-The-Large) or ITS (In-The-Small) models. Each model can be considered as system and process views, respectively, as follows:The ITL model is a system view that visually shows a set of processes in a system connected by a set of channels for communication among the processes;The ITS model is a process view that visually shows the set of interactions, that is, communication or movement, to perform in a sequential or selective order.

The internal components of the modeler are as follows:
The ITL loader loads each ITL model on the SAVE model view screen and shows all the processes and their communication channels, as defined on the ADOxx meta-modeling platform;The ITS loader loads each ITS model on the SAVE model view screen and shows all the actions and their precedence relations, as defined on the ADOxx meta-modeling platform;ITL/ITS mapper—as the system defined, an ITL model contains a number of ITS models. It inspects the proper relations between the ITL model and their contained ITS models for their proper syntactical relations and performs a function to generate a set of preliminary data that will be used to generate a set of future data for the following execution model for the system, including processes;T2M parser performs the translation function from the specification for system with processes in dTP-Calculus to ITL and ITS models in SAVE on ADOxx;Syntactic checker—At the time of checking the T2M parser function, it performs the inspection function for the syntactic validity of the characters input to the parser;The probability specifier supports a function to support the specification of probabilities in the ITS model.

#### 4.2.2. SAVE: Simulator

*Simulator* is one of the main tools in SAVE that generates a visual representation, namely, the *execution (EX) model*, of all the possible execution paths for a target ITL model and its included ITS models, including all the information related to geographical and temporal properties, namely, the *geo-temporal space (GTS) model*, with probabilities if needed, as follows:The execution (EX) model is a system transition model similar to reachability graph [16,17,18] that shows visually all the execution paths of a target ITL model with its included ITS models; its visual representation is as defined on the ADOxx meta-modeling platform for EX models;The geo-temporal space (GTS) model is a graphical model that shows the processes in a system and their interactions, that is, communication and movement, on a two-dimensional geo-temporal space, as defined on the ADOxx meta-modeling platform for GTS models.

The internal components of the simulator are as follows:
EM Generator generates an EX model for a target ITL model with a set of ITS models;Simulation Core with Probability is a core engine to simulate each path of the target EX model with probability and visually shows the probabilistic branches of the EM model for simulation;EM Path Analyzer performs a set of probabilistic analysis on each path of the target EX model;GTS Generator generates a GTS model from the EX model.

#### 4.2.3. SAVE: Analyzer and Verifier

*Analyzer and Verifier* (AV) is one of the mail tools in SAVE, which performs an analysis and verification of the *safety and security requirements* (SSReq) of the target system and generates the results of the analysis and verification visually using the GTS model of the target system.

The internal components of AV are as follows:T2G Logic Parser checks the syntactical validity of the safety and security requirements and performs a function to generate the preliminary data needed for the analysis and verification;GTS Logic Verifier performs an analysis and verification of whether the target GTS model is satisfied for the specified SSReq or not from the data received from the T2G Logic Parser;GTS Logic Visualizer performs a function to visualize the results of analysis and verification from the GTS Logic Verifier;Coverage Analyzer/Verifier performs a set of analyses and verifications of SSReqs on all the paths of the EX model for the target system, including visualization.

### 4.3. Smart EMS System

The EMS (emergency medical service) statistics from National Fire Department of Korea in 2021 [19] show the following:an 11.3% increase in transfer cases from 2020, which is 1,775,000 transfer cases;a 12.4% increase in transfer cases from 2020, which is 1,823,000 individual transferred patients.

Compared with those from 10 years ago, the increase rates of the cases and individual patients are 27.8% and 25.6%, respectively, which show that the rates have been increasing consistently.

Among the individual patients, the four major critical patient types are those with illnesses of the heart (cardiovascular disease), illnesses of the brain (cerebrovascular disease), cardiac arrest, and severe trauma, rating 44.2%, 39.2%, 11.4%, and 5.3%, respectively [19].

The smart EMS supported by smart IoT systems may be able to increase the survival rate of the critical patients dramatically if the proper EMS is provided.

EMS is the service utilizing the IoT technologies to enforce the activities of emergency medical workers responsible for on-spot emergency medical treatments and patient transfer at the time of emergency medical dispatch in order to increase the survival rate of the critical patients [20,21,22,23,24].

Compared with the existing EMS, smart EMS provides patients with IoT devices at the time of emergency medical workers arrive at the place of the emergency occurrence.

The medical information of the patients, including pulse, oxygen saturation, etc., is collected on the spot and transmitted to the 911 service and the proper medical institutes so that the proper treatments and prescriptions can be made by the medical doctors or specialists and, consequently, the golden time to rescue the patients’ lives can be obtained by the emergency medical teams at the time of the patients arrival at the medical institutes [25,26].

Such smart EMS based on the IoT is a futuristic EMS system where the biometrics of the patients, including electrocardiography, blood pressure, etc., are collected on the spot and transmitted to the proper medical institutes using a 5G network, their emergency medical situations are analyzed by the medical specialists in the institutes, the emergency medical workers are informed of the proper emergency medical pre-treatments for the patients by the specialists, and the most suitable medical institutes and the shortest and safest routes to the institutes are provided to the workers for the transfer of the patients.

#### 4.3.1. Description

Smart EMS (emergency medical service) systems deal with the emergency situations of critical patients with the support of smart IoT environments. When an emergency situation occurs with a critical patient, 911 is notified of the situation by some hotline, and 911 informs the ambulance of the situation for emergency medical treatment and transfer to the proper hospital if necessary.

In cases of multiple situations of this kind occurring simultaneously, it is necessary for 911 and ambulances to transfer the patients on the spot to proper hospitals based on some prediction with respect to the criticalities of the patients.

The critical status of the patients is assumed to be classified depending on emergency situations, as shown in Table 12.

The operational requirements specifications for smart EMS are as follows:(1)EMS consists of 911 (911Center_1_), ambulances (AmbX_1_, AmbY_1_), locations of patients (LocA, LocB, LocC, LocD), and hospitals (HospM_1_ and HospN_1_);
(i)911Center_1_ contains ambulances;(ii)Both AmbX_1_ and AmbY_1_ imply ambulances;(iii)LocA LocB, LocC and LocD imply locations;(iv)HospM_1_ and HospN_1_ imply hospitals.
(2)Assuming that there is a T1 patient of Table 12 in LocA;(3)Assuming that there is a T2 patient of Table 12 in LocB;(4)Assuming that there is a T2 patient of Table 12 in LocC;(5)Assuming that there is a T3 patient of Table 12 in LocD;(6)911Center1 receives patient information from locations LocA, LocB, LocC, and LocD;
(i)911Center1 sends AmbX1 both the priority information for rescue based on Table 12 and the patient information received from LocA and LocB;(ii)911Center1 sends AmbY1 both the priority information for rescue based on Table 12 and the patient information received from LocC and LocD.
(7)AmbX_1_ moves to LocA and LocB, based on the information received from 911Center_1_;
(i)In the case where AmbX_1_ moves to LocA first, it transfers the patient at the location to HospM_1_ after sending a message to the hospital and it moves to LocB, rescues the patient at the location, and transfers the patient to HospM_1_;(ii)In the case where AmbX_1_ moves to LocB first, instead of LocA, it transfers the patient at the location to HospM_1_ after sending a message to the hospital and it moves to LocA, rescues the patient at the location, and transfers the patient to HospM_1_.
(8)AmbY_1_ moves to LocC and LocD, based on the information received from 911Center_1_;
(i)In the case where AmbX_1_ moves to LocC first, it transfers the patient at the location to HospN_1_ after sending a message to the hospital and it moves to LocD, rescues the patient at the location, and transfers the patient to HospN_1_;(ii)In the case where AmbX_1_ moves to LocD first, instead of LocC, it transfers the patient at the location to HospN_1_ after sending a message to the hospital and it moves to LocC, rescues the patient at the location, and transfers the patient to HospN_1_.
(9)HospM_1_ takes a patient from AmbX_1_.(10)HospN_1_ takes a patient from AmbY_1_.

#### 4.3.2. Specification

Figure 7 shows the smart EMS example in dTP-Calculus. The detailed description of the dTP-Calculus code is as follows:
(1)$SmartEMS∷=LocA|LocB|LocC|LocD|{911Center}_{1}\left[{AmbX}_{1}|{AmbY}_{1}\right]|{HospM}_{1}|{HospN}_{1}$ shows the inclusion relations among the components of the EMS example;(2)LocA transmits the patient’s information to 911Center with $CALL911\_LocA\left(\bar{T1}\right)$;(3)LocB transmits the patient’s information to 911Center with $CALL911\_LocB\left(\bar{T2}\right)$;(4)LocC transmits the patient’s information to 911Center with $CALL911\_LocC\left(\bar{T2}\right)$;(5)LocD transmits the patient’s information to 911Center with $CALL911\_LocD\left(\bar{T3}\right)$;(6)911Center_1_ sends the patients’ information to AmbX_1_ and AmbY_1_;
(a)The probabilities that 911Center_1_ sends AmbX_1_ the rescue order for the patients are
(i)${ORDER\_AmbX}_{1}\left(\bar{T1-T2}\right)\left\{0.9\right\}$—the probability that a T1 patient at LocA is rescued first (90%);(ii)${ORDER\_AmbX}_{1}\left(\bar{T2-T1}\right)\left\{0.1\right\}$—the probability that a T2 patient at LocB is rescued first (10%).
(b)The probabilities that 911Center_1_ sends AmbY_1_ the rescue order for the patients are
(i)${ORDER\_AmbY}_{1}\left(\bar{T2-T3}\right)\left\{0.7\right\}$—the probability that a T2 patient at LocC is rescued first (70%);(ii)${ORDER\_AmbY}_{1}\left(\bar{T3-T2}\right)\left\{0.3\right\}$—the probability that a T2 patient at Location LocC is rescued first (30%).

(7)AmbX_1_ receives the patient’s information from 911Center_1_ and transfers the patient to HospM_1_ after rescuing the patient;
(a)${ORDER\_AmbX}_{1}\left(T1-T2\right)\left\{0.9\right\}$—the probability that AmbX_1_ moves first to LocA (90%);(b)${ORDER\_AmbX}_{1}\left(T2-T1\right)\left\{0.1\right\}$—the probability that AmbX_1_ moves first to LocB (10%);(c)${TRANSFER\_to\_HospM}_{1}\left(\bar{T1}\right)\left\{0.9\right\}$—the probability that AmbX_1_ transmits the information of a T1 patient at LocA to HospM_1_ (90%);(d)${TRANSFER\_to\_HospM}_{1}\left(\bar{T2}\right)\left\{0.1\right\}$—the probability that AmbX_1_ transmits the information of a T2 patient at LocB to HospM_1_ (10%).
(8)AmbY_1_ receives the patient’s information from 911Center_1_ and transfers the patient to HospN_1_ after rescuing the patient;
(a)${ORDER\_AmbY}_{1}\left(T2-T3\right)\left\{0.7\right\}$—the probability that AmbY_1_ moves first to LocC (70%);(b)${ORDER\_AmbY}_{1}\left(T3-T2\right)\left\{0.3\right\}$—the probability that AmbY_1_ moves first to LocD (30%);(c)${TRANSFER\_to\_HospN}_{1}\left(\bar{T2}\right)\left\{0.9\right\}$—the probability that AmbY_1_ transmits the information of a T2 patient at LocC to HospN_1_ (90%);(d)${TRANSFER\_to\_HospN}_{1}\left(\bar{T3}\right)\left\{0.1\right\}$—the probability that AmbY_1_ transmits the information of a T3 patient at LocD to HospN_1_ (10%).
(9)HospM_1_ receives the patient from AmbX_1_;
(a)${TRANSFER\_to\_HospM}_{1}\left(T1\right)\left\{0.9\right\}$—the probability that HospM_1_ receives the information on a T1 patient at LocA (90%);(b)${TRANSFER\_to\_HospM}_{1}\left(T2\right)\left\{0.1\right\}$—the probability that HospM_1_ receives the information on a T2 patient at LocB (10%).
(10)HospN_1_ receives the patient from AmbY_1_.
(a)${TRANSFER\_to\_HospN}_{1}\left(T2\right)\left\{0.9\right\}$—the probability that HospN_1_ receives the information on a T2 patient at LocC (90%);(b)${TRANSFER\_to\_HospN}_{1}\left(T3\right)\left\{0.1\right\}$—the probability that HospN_1_ receives the information on a T3 patient at LocD (10%).


**Figure 7 sensors-24-03881-f007:**
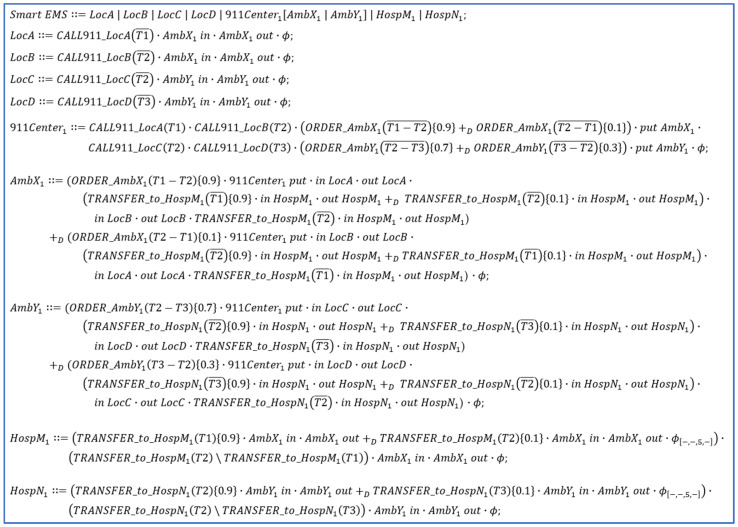
The smart EMS example in dTP-Calculus.

Figure 8 and Figure 9 show the ITL (In-The-Large) model and the ITS (In-The-Small) models in SAVE (Specification, Analysis, Verification, Evaluation) for the dTP-Calculus code in Figure 7.

#### 4.3.3. Probability Analysis

Figure 10 shows the EX (execution) model generated by the EX Model Generator in SAVE from the ITL model in Figure 8 and the ITS models in Figure 9 for the smart EMS example. The EX model shows 46 execution paths. The probabilities of all 46 paths can be generated for analysis by Path Analyzer in SAVE, as shown in Table 13. Its conceptual view can be depicted as shown in Figure 11, where each step of the paths implies the accumulated probabilities up to the steps of the paths. Note that the detailed views of the figures are in Appendix A.

#### 4.3.4. Safety and Security Requirements

Table 14 shows the safety and security requirements. In the table, $R^{Sc}$ implies security requirements where all the patients must be transferred to hospitals with a condition that the patient with a higher priority must be transferred prior to the patient with a lower priority. Similarly, in the table, $R^{Sf}$ implies security requirements where all the patients must be transferred before the deadline.

Figure 12 shows a GTS (geo-temporal space) model for the smart EMS example generated from SAVE. As stated, it represents the results of the simulation and verification performed on one of the execution paths for the smart EMS example. The GTS model shows the simulation view for the example by both processes with executing action blocks and their interacting arrows among the blocks of other processes as a background view, and a set of blue lines shows the successful results for the verification of the requirements for the specified $R^{Sc}$ and $R^{Sf}$. The detailed view is shown in Appendix A.

Table 15 shows the results for the validation of the probabilistic requirements for the 16 normal termination cases of the EMS example. It shows that only two Safety Requirements from the probabilistic requirements, that is, $R_{2}^{Sf}$ and $R_{3}^{Sf}$ are satisfied, as shown in Table 16 and Figure 13.

As shown in Figure 13 and Table 16, the present system cannot satisfy all the probabilistic safety and security requirements for the example. Note that the capabilities of the IoT for each version can be determined by those of the IoT devices with the probabilities, as shown in Table A1 and Table A2 in Appendix A.

In order to satisfy all the requirements, it is necessary to upgrade or enhance the performance of the related IoT devices in the system, as described in the previous section.

### 4.4. Smart EMS: Incremental System Improvement

As shown in Table 16, it was proved that all the probabilistic requirements for the example were not satisfiable due to a lack of full capabilities of the IoT devices in the system. In order to satisfy the requirements, it is necessary to improve the capability of the system incrementally with respect to those of the IoT devices in the system. This subsection will show how such an improvement can be achieved for the system using the notion of the system and process equivalences.

#### 4.4.1. Incremental System Improvement by Single Process Enhancement

Figure 14 shows the improvement of AmbX from AmbX_1_ to AmbX_4_ with the enhanced probabilities, as follows:
(1)The probabilities that 911Center_1_ sends AmbX_1_ the rescue order for the patients are updated where
${ORDER\_AmbX}_{1}\left(\bar{T1-T2}\right)\left\{0.99\right\}$—the probability that a T1 patient at LocA is rescued first (99%);${ORDER\_AmbX}_{1}\left(\bar{T2-T1}\right)\left\{0.01\right\}$—the probability that a T2 patient at LocB is rescued first (1%).
(2)AmbX_1_ receives the patient’s information from 911Center_1_, and transfers the patient to HospM_1_ after rescuing the patient, with the updated probabilities.
${ORDER\_AmbX}_{1}\left(T1-T2\right)\left\{0.99\right\}$—the probability that AmbX_1_ moves first to LocA (99%);${ORDER\_AmbX}_{1}\left(T2-T1\right)\left\{0.01\right\}$—the probability that AmbX_1_ moves first to LocB (1%).


**Figure 14 sensors-24-03881-f014:**
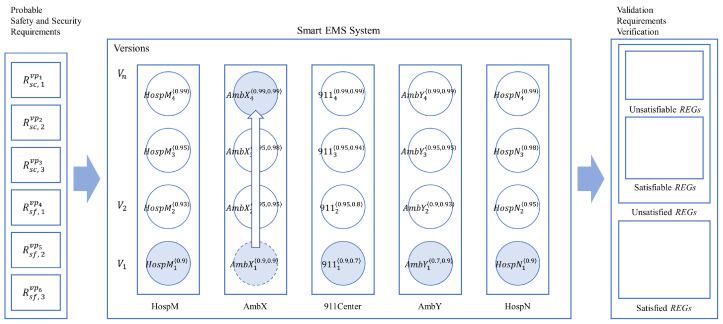
Improvement of AmbX for the smart EMS example: from AmbX_1_ to AmbX_4_.

Table 17 shows that the improvement is supported by the probabilistic equivalence between AmbX_1_ and AmbX_4_.

Figure 15 shows the EX model for the smart EMS system S_1_ (HospM_1_, AmbX_1_, 911Center_1_, AmbY_1_, HospN_1_) and the EX Model for the smart EMS system S_2_ (HospM_1_, AmbX_4_, 911Center_1_, AmbY_1_, HospN_1_) improved by the enhancement of AmbX_1_ to AmbX_4_.

The figure shows that the two EX models are identical in shape, but it is not clear whether they are probabilistically equivalent or not within the permissible range of probabilities.

In order to demonstrate probabilistic system equivalence between S_1_ and S_2_, Table 18 and Table 19 are constructed to represent the probabilities for the all the system transitions represented in the EX model for S_1_ and EX model for S_2_, respectively.

As shown in Table 18 and Table 19, all the probabilities in Table 18 are within the range of those in Table 19, that is, ($\Delta^{\langle \langle 0.0\rangle ,\langle 0.09,0.09\rangle ,\langle 0.0,0.0\rangle ,\langle 0.0,0.0\rangle ,\langle 0.0\rangle \rangle }$).

It implies that two systems are probabilistically equivalent within the ($\Delta^{\langle \langle 0.0\rangle ,\langle 0.09,0.09\rangle ,\langle 0.0,0.0\rangle ,\langle 0.0,0.0\rangle ,\langle 0.0\rangle \rangle }$) permissible range as follows:$${Smart\;EMS\;System\;S}_{1}\overset{\Delta^{\langle \langle 0.0\rangle ,\langle 0.09,0.09\rangle ,\langle 0.0,0.0\rangle ,\langle 0.0,0.0\rangle ,\langle 0.0\rangle \rangle }}{\approx }{Smart\;EMS\;System\;S}_{2}$$

Table 20 shows the validation results for probabilistic verification by the incremental improvement for Ambulance X from AmbX_1_ to AmbX_4_. The table shows that AmbX_4_ satisfies two more probabilistic requirements, that is, $R_{3}^{Sc}$ and $R_{1}^{Sf}$, than the previous system with the old version AmbX_1_. However, there are still two unsatisfied probabilistic security requirements, that is, $R_{1}^{Sc}$ and $R_{2}^{Sc}$, as verified in Table 21 and shown in Figure 16.

As shown in the above example, it is not guaranteed for all the safety and security requirements to be satisfied probabilistically through the improvement of a single IoT device. In addition, it is possible for the system to be upgraded with additional expenses for making a reasonable balance among IoT devices in the case of a preceding impractical improvement of a single IoT device. In other words, it could be more effective to make a well-balanced incremental improvement of multiple IoT devices.

#### 4.4.2. Incremental System Improvement for Collective Process Enhancements

Figure 17 shows the basic process equivalence relations for the process AmbX, AmbY, 911Center, HospM, and HospN for incremental improvement, as follows:(1)AmbX—assuming that AmbX has been upgraded one level up from AmbX_1_ to AmbX_2_;(2)911Center—assuming that 911Center related to AmbX has been upgraded two levels up from 911Center_1_ to 911Center_3_;(3)HospM—assuming that HospM has been upgraded two levels up from HospM_1_ to HospM_3_;(4)AmbY—assuming that AmbY has been upgraded two levels up from AmbY_1_ to AmbY_3_;(5)HospN—assuming that HospN has been upgraded one level up from HospN_1_ to HospN_2_.

**Figure 17 sensors-24-03881-f017:**
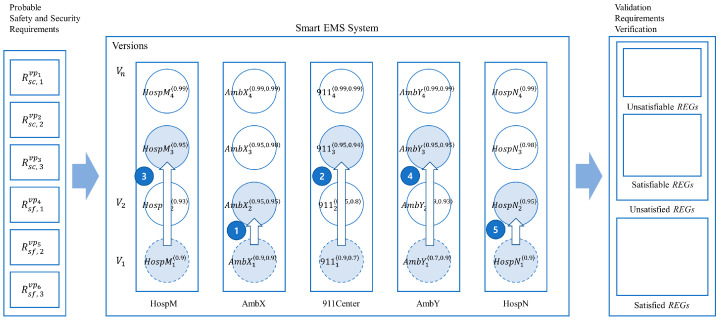
Multiple incremental improvements of the smart EMS example.

Table 22 shows a summary of the collective incremental improvements of the example based on the notion of probabilistic process equivalence, as demonstrated in Table 17 in the previous incremental improvement example.

As shown with the incremental improvement examples, it is noticeable that when some of the components in the system are improved probabilistically, the other components related to the improved components are subject to be improved accordingly while they are interacting with each other probabilistically.

Next, the relationship between the incremental improvement of the smart EMS example and the probabilistic requirements of the example is shown.

Figure 18 shows the EX model for the smart EMS system S_1_ (HospM_1_, AmbX_1_, 911Center_1_, AmbY_1_, HospN_1_) and EX model for its improved smart EMS system S_3_ (HospM_3_, AmbX_2_, 911Center_3_, AmbY_3_, HospN_2_) through the collective incremental enhancements generated by SAVE.

As described in the system improvement by single process enhancement, the two EX models are identical in shape, but it is not clear whether they are probabilistically equivalent or not within the permissible range of probabilities.

In order to demonstrate probabilistic system equivalence between S_1_ and S_3_, Table 23 and Table 24 are constructed to represent the probabilities of the all the system transitions represented in the EX model for S_1_ and EX model for S_3_, respectively.

As shown in Table 23 and Table 24, all the probabilities in Table 23 are within the range of those in Table 24, that is, ($\Delta^{\langle \langle 0.05\rangle ,\langle 0.05,0.05\rangle ,\langle 0.05,0.24\rangle ,\langle 0.25,0.05\rangle ,\langle 0.05\rangle \rangle }$). It implies that the two systems are probabilistically equivalent within the ($\Delta^{\langle \langle 0.05\rangle ,\langle 0.05,0.05\rangle ,\langle 0.05,0.24\rangle ,\langle 0.25,0.05\rangle ,\langle 0.05\rangle \rangle }$) permissible range as follows:$${Smart\;EMS\;System\;S}_{1}\overset{\Delta^{\langle \langle 0.05\rangle ,\langle 0.05,0.05\rangle ,\langle 0.05,0.24\rangle ,\langle 0.25,0.05\rangle ,\langle 0.05\rangle \rangle }}{\approx }{Smart\;EMS\;System\;S}_{3}$$

Table 25 shows the validation results for probabilistic requirements generated from the collective incremental improvements. In Table 26, it can be seen that all the probabilistic requirements are satisfied probabilistically based on the system probabilities. Additionally, the results of the validation of satisfiability are shown on the right side of Figure 19.

It is one of the advantages of the collective incremental improvement approach that the additional cost for the improvement can be minimized compared to the single improvement approach since the system can be improved based on the collective incremental improvements of strongly correlated IoT devices in the system. In addition, it can be more effective for the probabilistic system requirements to be satisfied at the system level than at its individual process levels.

## 5. Comparative Study

### 5.1. Previous Research

In previous research, we developed a methodology to specify smart IoT systems probabilistically with dTP-Calculus for their nondeterministic operational requirements and verify smart IoT systems with GTS logic for their safety and security requirements [11].

The previous research was motivated by the fact that the probabilistic process algebra for smart IoT systems is suitable to predict and control the nondeterministic behavior of the systems.

However, the existing probabilistic process algebra, such as PCCS (Probabilistic Calculus of Communicating Systems), PACSR (Probabilistic Algebra of Communicating Shared Resources), PALOMA (Process Algebra for Located Markovian Agents), are based only on static probabilistic models of unconditional nondeterministic choice operations, resulting a lack of probabilistic analytical methods for nondeterministic probabilistic behavior of the systems as follows [27,28,29]:PCCS only relies on discrete probability models;PACSR only relies on discrete probability models, similar to PCCS;PALOMA only focuses exponential probability distribution models.

In other words, they are not suitable to predict and control the nondeterministic behavior of the systems based on the dynamic probabilistic models.

In order to overcome the limitations, our previous research focused on developing a new process algebra, namely, dTP-Calculus, with dynamic probability property of a choice operation in order to specify the probabilistic operational requirements of smart IoT systems and to verify probabilistic safety and security requirements of the systems for the probabilistic operational requirements.

In this way, the nondeterministic behavior of the systems can be both predicted and controlled probabilistically in dynamically incremental manners. The target characteristics of the research for dTP-Calculus were as follows:Specification of nondeterministic behavior—the behavior of processes in a system is specified with probability so that the system become probabilistically predictable and controllable;Verification of probabilistic requirements—the safety and security requirements can be both analyzed and verified probabilistically;Control of nondeterministic behavior—the nondeterministic behavior of a system becomes controllable with dynamically controllable probability.

This approach shows that dTP-Calculus is a very effective method to specify, analyze, verify and to evaluate the requirements of smart IoT systems in smart cities.

Especially in case where the probabilistic operational requirements of the systems do not satisfy the probabilistic safety and security requirements of the systems, the probabilities for the operational requirements have to be modified to an arbitrary level so that the safety and security requirements can be satisfied probabilistically.

The previous research shows the feasibility of the approach to overcome the limitation, but the arbitrariness to find the degree of the probabilities to satisfy the requirements was not properly handled in the approach.

In order to handle the arbitrariness in the previous research, this paper develops the following two new notions of probabilistic equivalences and incremental improvement:Probabilistic equivalences—the probabilities for operational requirements on nondeterministic operators are defined by the incremental levels reflecting the capacities of the real IoT devices;Incremental improvement—the devices can be improved by incremental steps to satisfy the probabilistic safety and security requirements.

In this way, the arbitrariness limitation in the previous research is overcome. Table 27 shows the differences between the previous research and the one in this paper.

### 5.2. Related Research

The equivalence in process algebra shows whether two processes or systems behave bisimulatively. It implies that one process or system can be replaced with its equivalent process or system. Such a notion of equivalence can be used to analyze the characteristics of the systems in terms of process algebra: It is possible to validate whether a system is realized with respect to its design and vice versa by verifying equivalence between the design and the system [30,31,32,33].

In the case of PCCS, which is an extended version of CCS (Calculus of Communicating Systems) with probability, the equivalence is a very important notion for modeling and analyzing the target systems. It is used to check the characteristics of the systems by assuring identical execution paths with respect to the system states and their interactions. The equivalence notion of PCCS helps analyze and predict the behavior of the systems with its probabilistic models. That is, it is ensured that the various execution paths may have the same results and, consequently, their probability distributions are same [27,34].

In the case of PACSR, which is an extended version of ACSR (Algebra of Communicating Shared Resources) with probability, the equivalence is used to assure a number of characteristics of the given probabilistic models. It shows that different execution paths have the same probabilistic characteristics with respect to the system states and interactions, and, consequently, ensures the same system behavior [28,35].

The equivalences in both PCCS and PACSR have a continuous and holistic characteristic: equivalence over the continuous range of probability as a whole from the perspective of systems. Since they do not consider the equivalences at the level of their subordinate components, it could be necessary to replace the whole system in the case of some default situations, which is very cost-demanding. The probabilistic equivalences in the paper are defined for processes and systems, respectively. In order for two processes to be probabilistically equivalent, they should be identical, and the probabilities of identical choice operators must be in an acceptable incrementation level, reflecting the corresponding capacities of the target IoT device. In order for two systems to be probabilistically equivalent, they should be identical, and but the probabilities of identical compositional choice outcomes or paths must be in acceptable incrementation level, reflecting the corresponding capacities of the target IoT system.

In the case where the target system fails due to the failure of a specific IoT device, that is, the probabilistic operational requirements of the system do not satisfy the probabilistic safety and security requirements of the system, it is possible to upgrade the system with a more powerful device with an enhanced capability, based on the notion of probabilistic process equivalence, so that the upgraded system, based on the notion of probabilistic system equivalence, can satisfy the probabilistic safety and security requirements.

Table 28 shows the comparative differences between the research in this paper and others.

Other equivalence notions reported in current research are as follows:A linear process-algebraic format with data for probabilistic automata (Katoen, J. P., van de Pol, J., Stoelinga, M., & Timmer, M, 2012) that proposes an algebraic approach to verify bisimulation on data-dependent models and presents a method to inspect equivalence among processes based on data dependencies [36];Probabilistic bisimulation: Naturally on distributions (Hermanns, H., Krčál, J., & Křetínský, J, 2014) that deals with similar bisimulation from probabilistic process algebra and introduces a method to limit the size of state space in probabilistic process algebra and a method to reduce the state space effectively by applying the bisimulation [37].

However, they are not directly related to any verification or validation of the requirements for the systems.

## 6. Conclusions and Future Research

This paper presented a new methodology to predict and control uncertainty, which may cause risks or disasters, for smart IoT systems by defining the nondeterminism in the systems with probabilities of the choice operations in process algebra based on the notions of probabilistic process and system equivalences.

The methodology includes dTP-Calculus, the SAVE tool suite and the incremental improvement approach, which demonstrated that the uncertainty caused by nondeterminism in the system can be solved by upgrading the correlated processes, representing IoT devices with their equivalent ones within enhanced positive probabilities.

The paper demonstrates its feasibility with a smart EMS system as an example of smart IoT systems, which is one of the most practical applications for smart cities.

The methodology and approach can be considered some efficient practices in the area of process algebra to predict and control uncertainty and the risks caused by the nondeterministic behavior of smart IoT systems with probability.

The results of the research in this paper contribute to the manageability of nondeterministic uncertainty for smart IoT systems during their design and implementation phases of system development, as well as in the operating and maintenance phases. Especially, the results increase the reliability and safety of the systems by providing a systematic methodology to predict and control various risks that can occur in very complex systems like smart cities.

Future research may include the following:(1)Defining the notion of weighted integration of interrelated processes for incremental improvement—the approach in this paper assumes that all the processes in the system are considered to be candidates for improvement. However, it could be better to include only the ones influence by the main processes that cause the major and direct improvement of performance of the systems, based on some degree of weighted interrelationships. In this way, the redundant repetition of increments may be reduced dramatically by applying them simultaneously at once;(2)Systematic evaluation mechanism for the versionization of processes—in the approach, the capabilities of IoT devices are represented with versions with some discrete probabilities. In the future, we need some systematic way of representing their capabilities in terms of discrete probability;(3)Field application—currently, SAVE has been applied to conceptual model examples. A real example from the field of smart IoT systems needs to be selected to prove the feasibility of the approach in this paper with the SAVE tool suite.

Most importantly, SAVE is available as an open model tool in OMiLAB community [38] and can be applied to any open smart IoT system.

## Figures and Tables

**Figure 1 sensors-24-03881-f001:**
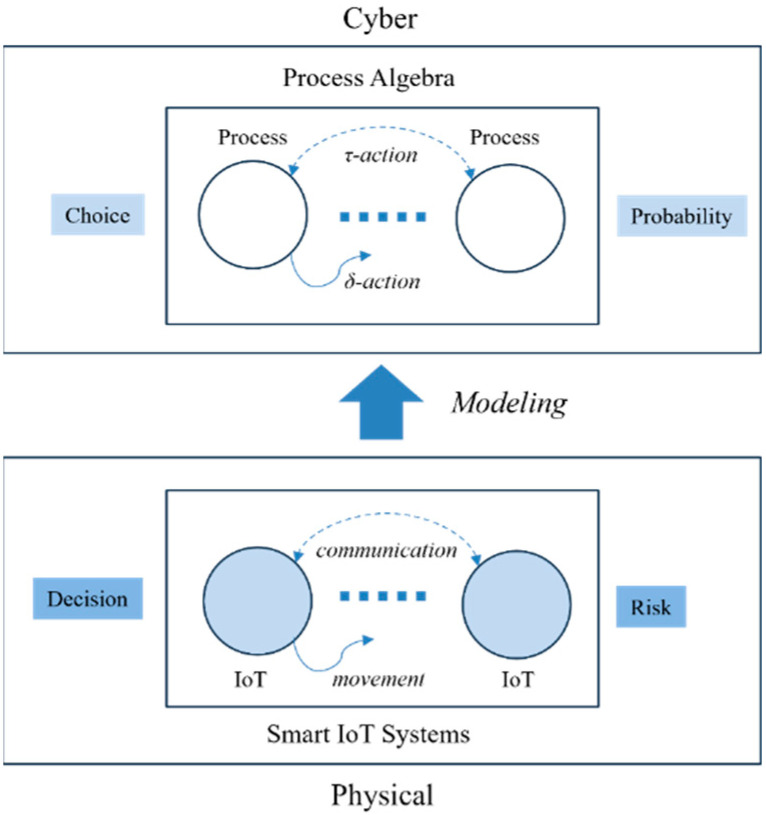
Process algebra for digital twins.

**Figure 3 sensors-24-03881-f003:**
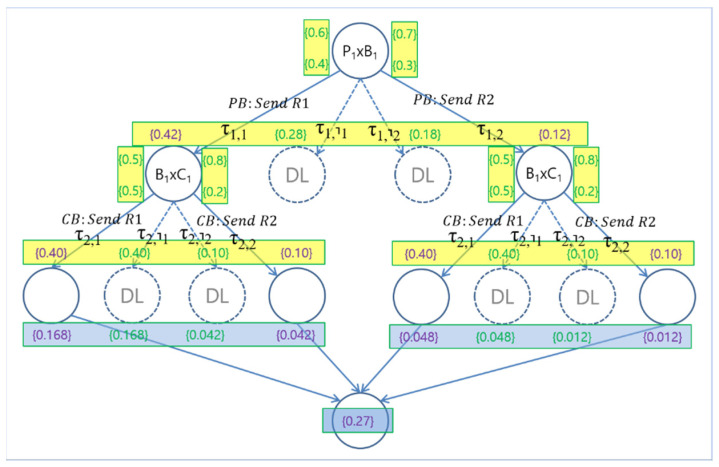
Conceptual probabilistic EX (execution) model.

**Figure 6 sensors-24-03881-f006:**
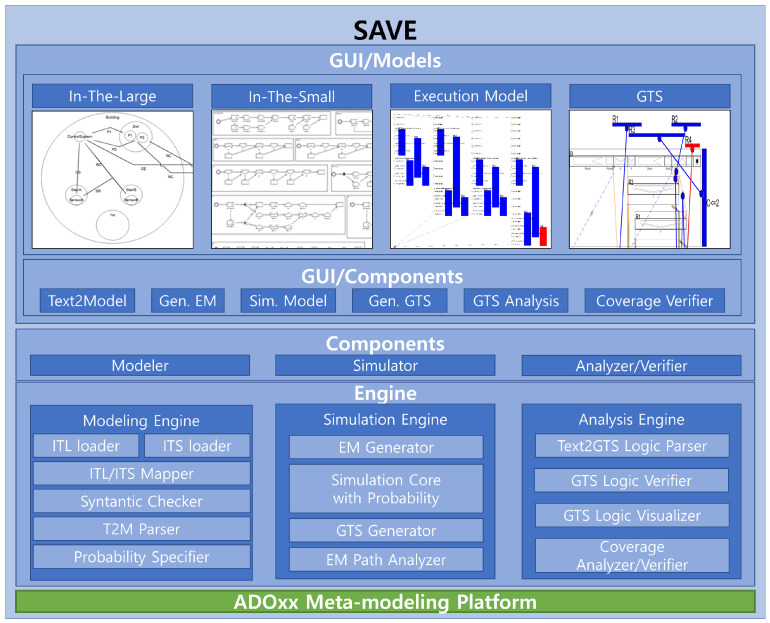
Components and architecture of SAVE.

**Figure 8 sensors-24-03881-f008:**
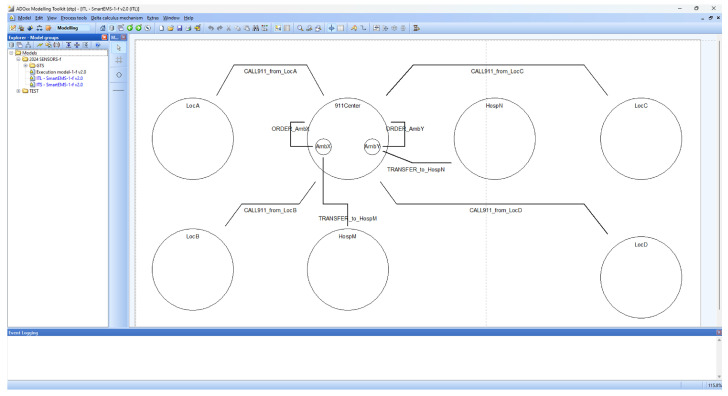
ITL model in SAVE for the smart EMS example (detailed view in Appendix A).

**Figure 9 sensors-24-03881-f009:**
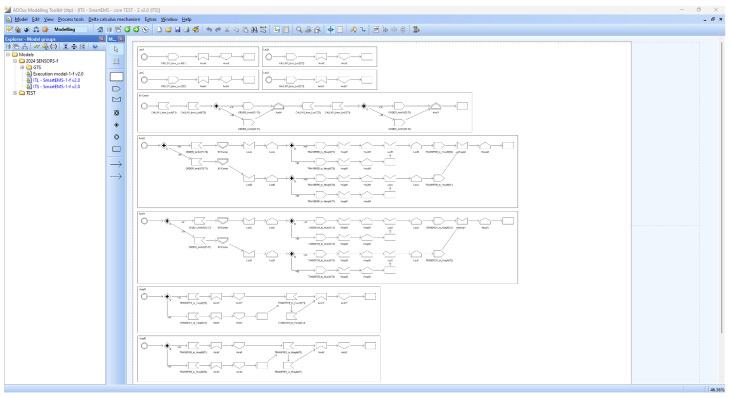
ITS models in SAVE for the smart EMS example (detailed view in Appendix A).

**Figure 10 sensors-24-03881-f010:**
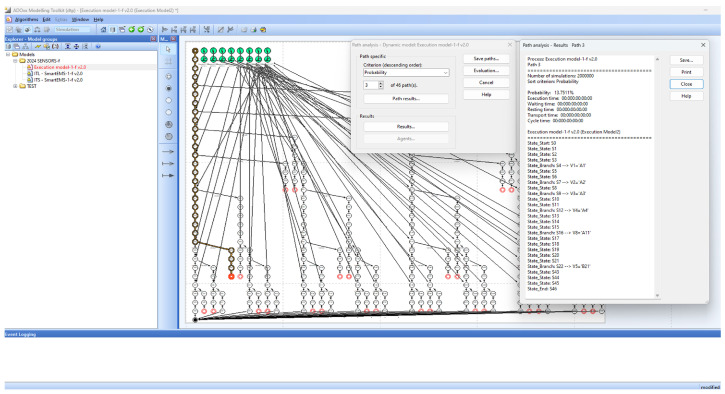
Path analysis of the EX model for the smart EMS example using SAVE (detailed view in Appendix A).

**Figure 11 sensors-24-03881-f011:**
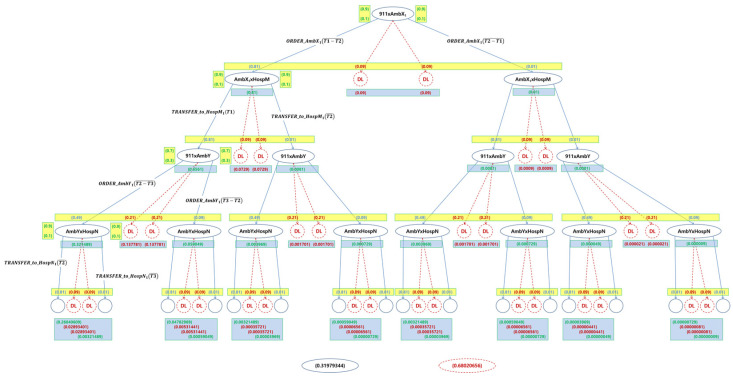
Conceptual representation of the step-wise accumulative probabilities for the smart EMS example (detailed view in Appendix A).

**Figure 12 sensors-24-03881-f012:**
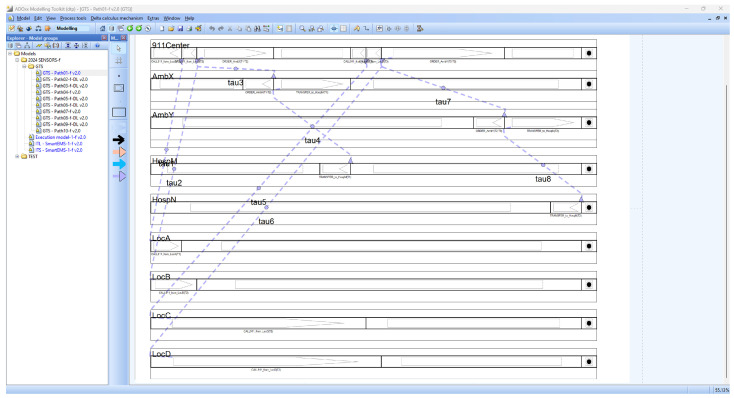
A GTS model for the smart EMS example using SAVE (detailed view in the Appendix A).

**Figure 13 sensors-24-03881-f013:**
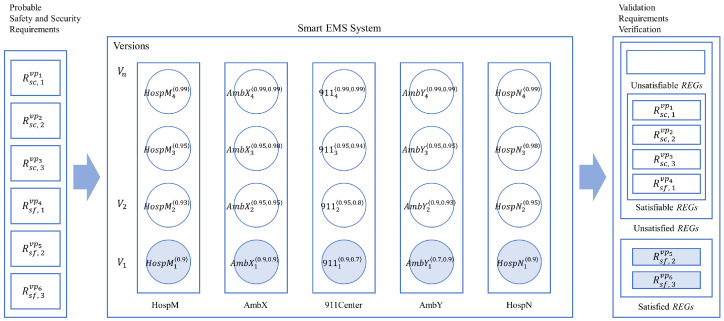
Analysis of the validation of the probabilistic requirements for the smart EMS example.

**Figure 15 sensors-24-03881-f015:**
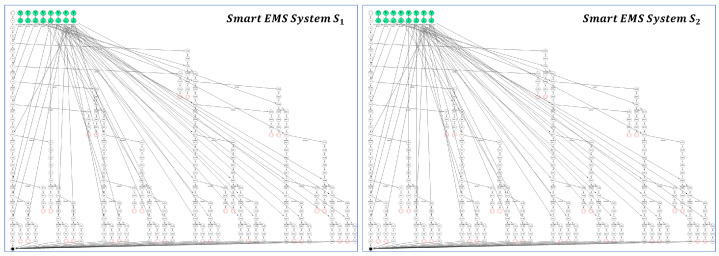
EX model for the smart EMS system S_1_ (HospM_1_, AmbX_1_, 911Center_1_, AmbY_1_, HospN_1_) and EX model for the smart EMS system S_2_ (HospM_1_, AmbX_4_, 911Center_1_, AmbY_1_, HospN_1_) for incremental improvement by single process enhancement.

**Figure 16 sensors-24-03881-f016:**
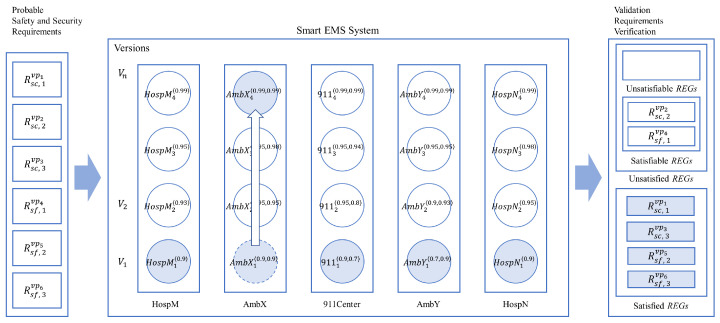
Analysis of the validation of probabilistic requirements for the smart EMS example: from AmbX_1_ to AmbX_4_.

**Figure 18 sensors-24-03881-f018:**
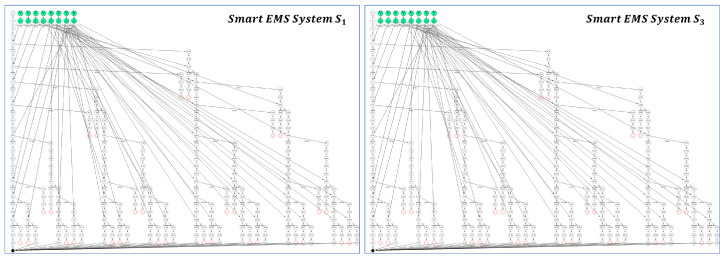
EX model for the smart EMS system S_1_ (HospM_1_, AmbX_1_, 911Center_1_, AmbY_1_, HospN_1_) and EX model for its improved smart EMS system S_3_ (HospM_3_, AmbX_2_, 911Center_3_, AmbY_3_, HospN_2_) for collective incremental enhancements.

**Figure 19 sensors-24-03881-f019:**
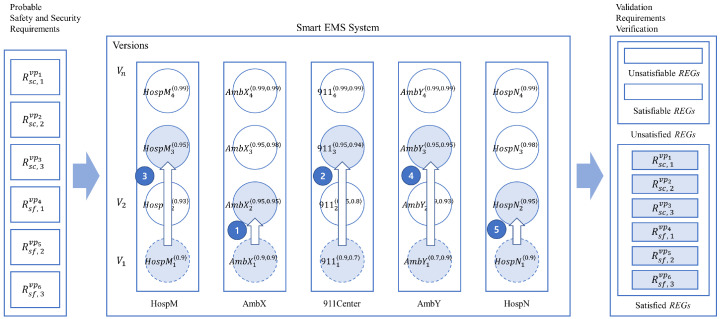
Analysis of the validation of probabilistic requirements for the smart EMS example through multiple incremental improvements.

**Table 2 sensors-24-03881-t002:** Synchronicity for movements in dTP-Calculus.

Movements	Request	Permision	Results
*In*	A=in B	B=A in	A in B
	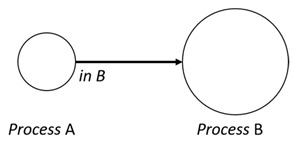	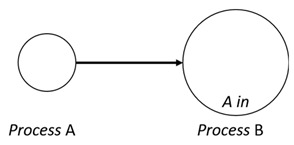	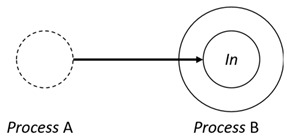
*Out*	A=out B	B=A out	A out B
	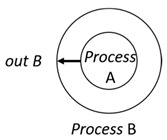	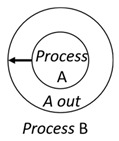	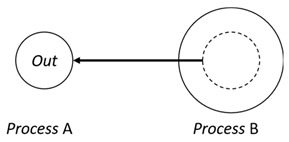
*Get*	B=get A	A=B get	B get A
	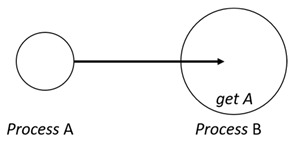	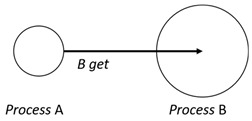	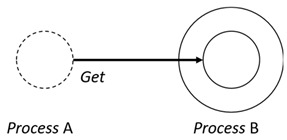
*Put*	B=put A	A=B put	B put A
	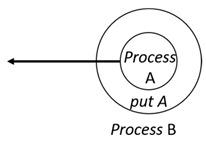	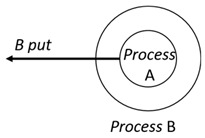	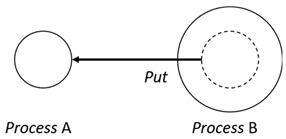

**Table 3 sensors-24-03881-t003:** dTP-Calculus syntax.

Construct		Name	Index
P=	A	Action	(1)
|	$A_{\left[r,to,e,d\right]}^{per,\;\;\;n}$	Timed Action	(2)
|	$P_{\left[r,to,e,d\right]}^{per,\;\;\;n}$	Timed Process	(3)
|	$P_{\left(pri\_n\right)}$	Priority	(4)
|	$P\left[Q\right]$	Nesting	(5)
|	$P\left\langle ch\right\rangle$	Channel	(6)
|	P+Q	Choice	(7)
|	Ppc+QFpc	Probabilistic Choice	(8)
|	P∥Q	Parallel	(9)
|	P\E	Exception	(10)
|	A.P	Sequence	(11)
F=	D	Discrete Distribution	(12)
|	$N\left(\mu ,\sigma \right)$	Normal Distribution	(13)
|	$Ex\left(\lambda \right)$	Exponential Distribution	(14)
|	$U\left(l,u\right)$	Uniform Distribution	(15)
A=	ϕ	Empty	(16)
|	$ch\left(\bar{msg}\right)$	Send	(17)
|	$ch\left(msg\right)$	Receive	(18)
|	M	Movement	(19)
|	C	Control	(20)
M=	$m^{pri}(k)\;P$	Movement Request	(21)
|	P m(k)	Movement Permission	(22)
m=	in	In Movement	(23)
|	get	Out Movement	(24)
|	out	Get Movement	(25)
|	put	Put Movement	(26)
C=	new P	Create Process	(27)
|	kill P	Kill Process	(28)
|	exit	Exit Process	(29)

**Table 4 sensors-24-03881-t004:** dTP-Calculus semantics.

Name	Transition Rules	Index
*Sequence*	$\frac{-}{A\cdot P\overset{\;A\;}{\to }P}$	(1)
*ChoiceL* *ChoiceR*	$\frac{-}{P+Q\to P}$, $\frac{-}{P+Q\to Q}$	(2)
*Prob.* *Choice*	$\frac{A\cdot P\overset{A}{\to }\;P}{(\sum_{i\in I}A_{i}\{{pc}_{i}\})\cdot P\overset{A_{i}\{{pc}_{i}\}}{\to }P\prime }(\sum_{i\in I}{pc}_{i}=1,\;i\in I)$	(3)
*ParallelL* *ParallelR*	$\frac{P\to P\prime }{P||Q\to P\prime ||Q}$, $\frac{Q\to Q\prime }{P||Q\to P||Q\prime }$	(4)
*ParallelCom*	$\frac{P\overset{\;A\;}{\to }P\prime ,Q\overset{\;\bar{A}\;}{\to }Q\prime }{P||Q\overset{\;\tau \;}{\to }P\prime ||Q\prime }$	(5)
*NestingO* *NestingI*	$\frac{P\to P\prime }{P[Q]\to P\prime [Q]}$, $\frac{Q\to Q\prime }{P[Q]\to P[Q\prime ]}$	(6)
*NestingCom*	$\frac{P\overset{\;A\;}{\to }P\prime ,Q\overset{\;\bar{A}\;}{\to }Q\prime }{P||Q\overset{\;\tau \;}{\to }P\prime ||Q\prime }$	(7)
*In*	$\frac{P\overset{\;in\left(k\right)Q}{\to }P\prime ,Q\overset{\;Pin\left(k\right)}{\to }Q\prime }{P||Q\overset{\;\delta \;}{\to }Q\prime [P\prime ]}$	(8)
*Out*	$\frac{P\overset{\;out\left(k\right)Q}{\to }P\prime ,Q\overset{\;Pout\left(k\right)}{\to }Q\prime }{Q[P]\overset{\;\delta \;}{\to }P\prime ||Q\prime }$	(9)
*Get*	$\frac{P\overset{\;get\left(k\right)Q}{\to }P\prime ,Q\overset{\;Pget\left(k\right)}{\to }Q\prime }{P||Q\overset{\;\delta \;}{\to }P\prime [Q\prime ]}$	(10)
*Put*	$\frac{P\overset{\;put\left(k\right)Q}{\to }P\prime ,Q\overset{\;Pput\left(k\right)}{\to }Q\prime }{P[Q]\overset{\;\delta \;}{\to }P\prime ||Q\prime }$	(11)
*InP*	$\frac{P\overset{\;in^{pri}\left(k\right)Q}{\to }P\prime }{P_{\left(n_{1}\right)}||Q_{\left(n_{2}\right)}\overset{\;\delta \;}{\to }Q_{\left(n_{2}\right)}[P\prime_{\left(n_{1}\right)}]}((n_{1}>n_{2}\wedge n_{2}\neq 0)\vee (n_{1}=0\wedge n_{2}\neq 0))$	(12)
*OutP*	$\frac{P\overset{\;out^{pri}\left(k\right)Q}{\to }P\prime }{Q_{\left(n_{2}\right)}[P_{\left(n_{1}\right)}]\overset{\;\delta \;}{\to }P\prime_{\left(n_{1}\right)}||Q_{\left(n_{2}\right)}}((n_{1}>n_{2}\wedge n_{2}\neq 0)\vee (n_{1}=0\wedge n_{2}\neq 0))$	(13)
*GetP*	$\frac{P\overset{\;get^{pri}\left(k\right)Q}{\to }P\prime }{P_{\left(n_{1}\right)}||Q_{\left(n_{2}\right)}\overset{\;\delta \;}{\to }P\prime_{\left(n_{1}\right)}[Q_{\left(n_{2}\right)}]}((n_{1}>n_{2}\wedge n_{2}\neq 0)\vee (n_{1}=0\wedge n_{2}\neq 0))$	(14)
*PutP*	$\frac{P\overset{\;put^{pri}\left(k\right)Q}{\to }P\prime }{P_{\left(n_{1}\right)}[Q_{\left(n_{2}\right)}]\overset{\;\delta \;}{\to }P\prime_{\left(n_{1}\right)}||Q_{\left(n_{2}\right)}}((n_{1}>n_{2}\wedge n_{2}\neq 0)\vee (n_{1}=0\wedge n_{2}\neq 0))$	(15)
*TickTime* *R*	$\frac{-}{A_{\left[r,to,e,d\right]}^{per,\;\;\;n}\cdot P\overset{\;⊳1\;}{\to }A_{\left[r-1,to,e,d-1\right]}^{per,\;\;\;n}\cdot P}(r\geq 1)$	(16)
*TickTime* *TO*	$\frac{A\cdot P||\bar{A}\cdot Q\overset{\;\tau \vee \delta \;}{\to }P||Q}{A_{\left[0,to,e,d\right]}^{per,\;\;\;n}\cdot P\overset{\;⊳1\;}{\to }A_{\left[0,to-1,e,d-1\right]}^{per,\;\;\;n}\cdot P}(to\geq 1)$	(17)
*TickTime* *End*	$\frac{-}{A_{\left[0,to,0,d\right]}^{per,\;\;\;n}\cdot P\overset{\;⊳1\;}{\to }P}$	(18)
*TickTime* *SyncE*	$\frac{A\cdot P||\bar{A}\cdot Q\overset{\;\tau \vee \delta \;}{\to }P||Q}{A_{\left[0,to_{1},e_{1},d_{1}\right]}^{{per}_{1},\;\;\;n_{1}}\cdot P||{\bar{A}}_{\left[0,to_{2},e_{2},d_{2}\right]}^{{per}_{2},\;\;\;n_{2}}\cdot Q\overset{\;⊳1\;}{\to }A_{\left[0,to_{1},e_{1}-1,d_{1}-1\right]}^{{per}_{1},\;\;\;n_{1}}\cdot P||{\bar{A}}_{\left[0,to_{2},e_{2}-1,d_{2}-1\right]}^{{per}_{2},\;\;\;n_{2}}\cdot Q}(e_{1}\geq 1,e_{2}\geq 1)$	(19)
*TickTime* *AsyncE*	$\frac{-}{A_{\left[0,to,e,d\right]}^{per,\;\;\;n}\cdot P\overset{\;⊳1\;}{\to }A_{\left[0,to,e-1,d-1\right]}^{per,\;\;\;n}\cdot P}(e\geq 1)$	(20)
*Timeout*	$\frac{-}{(A_{\left[0,0,e,d\right]}^{per,\;\;\;n}\backslash E)\cdot P\overset{\;⊳1\;}{\to }E\cdot P}$	(21)
*Deadline*	$\frac{-}{(A_{\left[r,to,e,0\right]}^{per,\;\;\;n}\backslash E)\cdot P\overset{\;⊳1\;}{\to }E\cdot P}$	(22)
*Period*	$\frac{-}{A_{\left[r,to,e,d\right]}^{per,\;\;\;n}\cdot P\overset{\;⊳per\;}{\to }A_{\left[r,to,e,d\right]}^{per,\;\;\;n-1}\cdot P}(n\geq 1)$	(23)
*Period End*	$\frac{-}{A_{\left[0,to,0,d\right]}^{per,0}\cdot P\overset{\;⊳1\;}{\to }P}$	(24)

**Table 6 sensors-24-03881-t006:** SSReqs for the PBC (Producer–Buffer–Consumer) example.

SSReq			VaReg	Validation
$R_{1}^{ss}$	$R_{1}^{sc}$	The order of the resources, that is, *R*1–*R*2 or *R*2–*R*1, should not be violated because the security information is contained in the first resource (*R*1) to decode the second resource (*R*2) or vice versa.	0.10	To be Validated
	$R_{2}^{sc}$	The propagation between the first (*R*1) and second (*R*2), or vice versa, should not be more than 3 time units.	0.15	
$R_{2}^{ss}$	$R_{1}^{sf}$	The deadline for the consumption of the second resource (*R*2) by $C_{1}^{\prime }$ should not be more than 10 time units for the order of the resources *R*1–*R*2.	0.10	To be Validated
	$R_{2}^{sf}$	The deadline for consumption of the second resource (*R*2) by $C_{1}^{\prime }$ should not be more than 11 time units for the order of the resources *R*2–*R*1.	0.05	

**Table 8 sensors-24-03881-t008:** Results of the validation of SSReqs: Case 1.

SSReq		Case 1		$P_{1}^{\prime }B_{1}^{\prime }C_{1}^{\prime }$
$R_{1}^{SS}$	$R_{1}^{sc}$	0.10	Satisfied	0.18
	$R_{2}^{sc}$	0.15	Satisfied	0.27
$R_{2}^{SS}$	$R_{1}^{sf}$	0.10	Satisfied	0.18
	$R_{2}^{sf}$	0.05	Satisfied	0.09

**Table 9 sensors-24-03881-t009:** Results of validation for SSReqs: Case 2.

SSReq		Case 2		$P_{1}^{\prime }B_{1}^{\prime }C_{1}^{\prime }$
$R_{1}^{SS}$	$R_{1}^{sc}$	0.20	Unsatisfied	0.18
	$R_{2}^{sc}$	0.25	Satisfied	0.27
$R_{2}^{SS}$	$R_{1}^{sf}$	0.20	Unsatisfied	0.18
	$R_{2}^{sf}$	0.05	Satisfied	0.09

**Table 11 sensors-24-03881-t011:** Results of validation for SSReqs: Case 2 (improved).

SSReq		Case 2		$P_{1}^{\prime }B_{1}^{\prime }C_{1}^{\prime }$	$P_{2}^{\prime }B_{1}^{\prime }C_{1}^{\prime }$
$R_{1}^{SS}$	$R_{1}^{sc}$	0.20	Unsatisfied, but Satisfiable	0.18	0.205
	$R_{2}^{sc}$	0.25	Satisfied	0.27	0.290
$R_{2}^{SS}$	$R_{1}^{sf}$	0.20	Unsatisfied, but Satisfiable	0.18	0.205
	$R_{2}^{sf}$	0.05	Satisfied	0.09	0.085

**Table 12 sensors-24-03881-t012:** Emergency classification for patients.

Status	Meaning	Example
T1	Immediate	1st Transfer Priority: Patient whose life is threatened if not treated meediately (e.g., heart attack, ceretral hemorrhage, major amputation, etc.).
T2	Delayed	2nd Transfer Priority: Patient who is under observation for immediate medical treatment if necessary, but not needed immediately (e.g., fracture, dislocation, food poisoning, etc.).
T3	Minimal	3rd Transfer Priority: Patient without any life-threaning or physical disabilities (e.g., minor lacerated wound, sprain, scratch, etc.).

**Table 13 sensors-24-03881-t013:** Path probabilities for the smart EMS example using SAVE.

Path	Probability	Path	Probability
1	0.260514	24 (Deadlock)	0.090095
2 (Deadlock)	0.028985	25	0.003193
3 (Deadlock)	0.028968	26 (Deadlock)	0.000362
4	0.003227	27 (Deadlock)	0.000355
5 (Deadlock)	0.137511	28	0.000042
6 (Deadlock)	0.137929	29 (Deadlock)	0.001706
7	0.047687	30 (Deadlock)	0.001679
8 (Deadlock)	0.005240	31	0.000601
9 (Deadlock)	0.005317	32 (Deadlock)	0.000071
10	0.000629	33 (Deadlock)	0.000073
11 (Deadlock)	0.072801	34	0.000008
12 (Deadlock)	0.073120	35 (Deadlock)	0.000840
13	0.003201	36 (Deadlock)	0.000924
14 (Deadlock)	0.000347	37	0.000037
15 (Deadlock)	0.000358	38 (Deadlock)	0.000005
16	0.000034	39 (Deadlock)	0.000005
17 (Deadlock)	0.001706	40	0.000002
18 (Deadlock)	0.001695	41 (Deadlock)	0.000021
19	0.000589	42 (Deadlock)	0.000019
20 (Deadlock)	0.000062	43	0.000008
21 (Deadlock)	0.000065	44 (Deadlock)	0.000001
22	0.000005	45 (Deadlock)	0.000002
23 (Deadlock)	0.089958	46	0.000003
Total			1.000000

**Table 14 sensors-24-03881-t014:** SSReqs (safety and security requirements) for the smart EMS example.

Security		Requirements	VaProb
$R^{Sc}$	$R_{1}^{Sc}$	All patients are to be transferred to hospitals.	0.35
	$R_{2}^{Sc}$	A T1 patient should be transferred before a T2 patient.	0.40
	$R_{3}^{Sc}$	A T2 patient should be transferred before a T3 patient.	0.30
Safety			
$R^{Sf}$	$R_{1}^{Sf}$	The deadline for a T1 patient is in 10 unit times.	0.50
	$R_{2}^{Sf}$	The deadline for a T2 patient is in 20 unit times.	0.30
	$R_{3}^{Sf}$	The deadline for a T3 patient is in 30 unit times.	0.20

**Table 15 sensors-24-03881-t015:** Analysis of the probabilistic verification of the smart EMS example.

Path		1	2	3	4	5	6	7	8	9	10	11	12	13	14	15	16	17	18	19	20	21	22	23	24	25	26	27	28	29	30	31	32	33	34	35	36	37	38	39	40	41	42	43	44	45	46	Prob.
Prob.	Succ.	0.26			0.00			0.06			0.00			0.00			0.00			0.00			0.00			0.00			0.00			0.00			0.00			0.00			0.00			0.00			0.00	0.32
	Fail		0.03	0.03		0.14	0.14		0.01	0.01		0.07	0.07		0.00	0.00		0.00	0.00		0.00	0.00		0.09	0.09		0.00	0.00		0.00	0.00		0.00	0.00		0.00	0.00		0.00	0.00		0.00	0.00		0.00	0.00		0.68
$R^{Sc}$	$R_{1}^{Sc}$	O	X	X	O	X	X	O	X	X	O	X	X	O	X	X	O	X	X	O	X	X	O	X	X	O	X	X	O	X	X	O	X	X	O	X	X	O	X	X	O	X	X	O	X	X	O	0.32
	$R_{2}^{Sc}$	O	X	X	O	X	X	O	X	X	O	X	X	O	X	X	O	X	X	O	X	X	O	X	X	X	X	X	X	X	X	X	X	X	X	X	X	X	X	X	X	X	X	X	X	X	X	0.32
	$R_{3}^{Sc}$	O	X	X	O	X	X	X	X	X	X	X	X	O	X	X	O	X	X	X	X	X	X	X	X	O	X	X	O	X	X	X	X	X	X	X	X	O	X	X	O	X	X	X	X	X	X	0.27
$R^{Sf}$	$R_{1}^{Sf}$	O	X	X	O	X	X	O	X	X	O	X	X	X	X	X	X	X	X	X	X	X	X	X	X	X	X	X	X	X	X	X	X	X	X	X	X	X	X	X	X	X	X	X	X	X	X	0.31
	$R_{2}^{Sf}$	O	X	X	O	X	X	O	X	X	O	X	X	O	X	X	O	X	X	O	X	X	O	X	X	O	X	X	O	X	X	O	X	X	O	X	X	X	X	X	X	X	X	X	X	X	X	0.32
	$R_{3}^{Sf}$	O	X	X	O	X	X	O	X	X	O	X	X	O	X	X	O	X	X	O	X	X	O	X	X	O	X	X	O	X	X	O	X	X	O	X	X	O	X	X	O	X	X	O	X	X	O	0.32

**Table 16 sensors-24-03881-t016:** Analysis of the validation of the probabilistic requirements for the smart EMS example.

Requirements		Va Probabilities	System Probabilities	Verification Results	
$R^{Sc}$	$R_{1}^{Sc}$	0.35	0.32	X	Unsatisfied, but Satisfiable
	$R_{2}^{Sc}$	0.40	0.32	X	Unsatisfied, but Satisfiable
	$R_{3}^{Sc}$	0.30	0.27	X	Unsatisfied, but Satisfiable
$R^{Sf}$	$R_{1}^{Sf}$	0.50	0.31	X	Unsatisfied, but Satisfiable
	$R_{2}^{Sf}$	0.30	0.32	O	Satisfied
	$R_{3}^{Sf}$	0.20	0.32	O	Satisfied

**Table 17 sensors-24-03881-t017:** Probabilistic equivalence between AmbX_1_ and AmbX_4_.

Versions	dTP-Calculus
AmbX	AmbX_1_	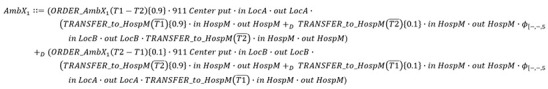
AmbX_4_	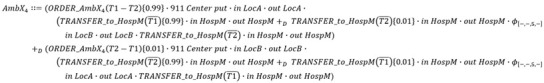
Probabilistic Process Equivalence	AmbX10.9,0.9~Δ0.09,0.09AmbX40.99,0.99

**Table 18 sensors-24-03881-t018:** Probabilistic reachability table for the smart EMS system S_1_(HospM_1_, AmbX_1_, 911Center_1_, AmbY_1_, HospN_1_).

BR	Reachability Table																																													
1	0.9																							0.1																						
2	0.9																						0.1	0.9	0.1																					
3	0.9											0.1													0.9											0.1										
4	0.9										0.1	0.9	0.1												0.9										0.1	0.9	0.1									
5	0.7					0.3							0.7					0.3							0.7					0.3							0.7					0.3				
6	0.7				0.3	0.7	0.3						0.7				0.3	0.7	0.3						0.7				0.3	0.7	0.3						0.7				0.3	0.7	0.3			
7	0.9		0.1				0.9		0.1				0.9		0.1				0.9		0.1				0.9		0.1				0.9		0.1				0.9		0.1				0.9		0.1	
8	0.9	0.1	0.9	0.1			0.9	0.1	0.9	0.1			0.9	0.1	0.9	0.1			0.9	0.1	0.9	0.1			0.9	0.1	0.9	0.1			0.9	0.1	0.9	0.1			0.9	0.1	0.9	0.1			0.9	0.1	0.9	0.1
Path	1	2	3	4	5	6	7	8	9	10	11	12	13	14	15	16	17	18	19	20	21	22	23	24	25	26	27	28	29	30	31	32	33	34	35	36	37	38	39	40	41	42	43	44	45	46
Prob.	0.26	0.03	0.03	0.00	0.14	0.14	0.06	0.01	0.01	0.00	0.07	0.07	0.00	0.00	0.00	0.00	0.00	0.00	0.00	0.00	0.00	0.00	0.09	0.09	0.00	0.00	0.00	0.00	0.00	0.00	0.00	0.00	0.00	0.00	0.00	0.00	0.00	0.00	0.00	0.00	0.00	0.00	0.00	0.00	0.00	0.00
Total	0.32 (Success) + 0.68 (Fail) = 1																																													

The background color in this table indicates deadlock cases.

**Table 19 sensors-24-03881-t019:** Probabilistic reachability table for the smart EMS system S_2_ (HospM_1_, AmbX_4_, 911Center_1_, AmbY_1_, HospN_1_).

BR	Reachability Table																																													
1	0.9																							0.1																						
2	0.99																						0.01	0.99	0.01																					
3	0.99											0.01													0.99											0.01										
4	0.9										0.1	0.9	0.1												0.9										0.1	0.9	0.1									
5	0.7					0.3							0.7					0.3							0.7					0.3							0.7					0.3				
6	0.7				0.3	0.7	0.3						0.7				0.3	0.7	0.3						0.7				0.3	0.7	0.3						0.7				0.3	0.7	0.3			
7	0.9		0.1				0.9		0.1				0.9		0.1				0.9		0.1				0.9		0.1				0.9		0.1				0.9		0.1				0.9		0.1	
8	0.9	0.1	0.9	0.1			0.9	0.1	0.9	0.1			0.9	0.1	0.9	0.1			0.9	0.1	0.9	0.1			0.9	0.1	0.9	0.1			0.9	0.1	0.9	0.1			0.9	0.1	0.9	0.1			0.9	0.1	0.9	0.1
Path	1	2	3	4	5	6	7	8	9	10	11	12	13	14	15	16	17	18	19	20	21	22	23	24	25	26	27	28	29	30	31	32	33	34	35	36	37	38	39	40	41	42	43	44	45	46
Prob.	0.32	0.04	0.04	0.00	0.16	0.16	0.06	0.01	0.01	0.00	0.09	0.09	0.00	0.00	0.00	0.00	0.00	0.00	0.00	0.00	0.00	0.00	0.01	0.01	0.00	0.00	0.00	0.00	0.00	0.00	0.00	0.00	0.00	0.00	0.00	0.00	0.00	0.00	0.00	0.00	0.00	0.00	0.00	0.00	0.00	0.00
Total	0.38 (Success) + 0.62 (Fail) = 1																																													

The background color in this table indicates deadlock cases.

**Table 20 sensors-24-03881-t020:** Analysis of system probability for the smart EMS example (improvement to AmbX_4_).

Path		1	2	3	4	5	6	7	8	9	10	11	12	13	14	15	16	17	18	19	20	21	22	23	24	25	26	27	28	29	30	31	32	33	34	35	36	37	38	39	40	41	42	43	44	45	46	Prob.
Prob.	Succ.	0.32			0.00			0.06			0.00			0.00			0.00			0.00			0.00			0.00			0.00			0.00			0.00			0.00			0.00			0.00			0.00	0.38
	Fail		0.04	0.04		0.16	0.16		0.01	0.01		0.09	0.09		0.00	0.00		0.00	0.00		0.00	0.00		0.01	0.01		0.00	0.00		0.00	0.00		0.00	0.00		0.00	0.00		0.00	0.00		0.00	0.00		0.00	0.00		0.62
$R^{Sc}$	$R_{1}^{Sc}$	O	X	X	O	X	X	O	X	X	O	X	X	O	X	X	O	X	X	O	X	X	O	X	X	O	X	X	O	X	X	O	X	X	O	X	X	O	X	X	O	X	X	O	X	X	O	0.38
	$R_{2}^{Sc}$	O	X	X	O	X	X	O	X	X	O	X	X	O	X	X	O	X	X	O	X	X	O	X	X	X	X	X	X	X	X	X	X	X	X	X	X	X	X	X	X	X	X	X	X	X	X	0.38
	$R_{3}^{Sc}$	O	X	X	O	X	X	X	X	X	X	X	X	O	X	X	O	X	X	X	X	X	X	X	X	O	X	X	O	X	X	X	X	X	X	X	X	O	X	X	O	X	X	X	X	X	X	0.32
$R^{Sf}$	$R_{1}^{Sf}$	O	X	X	O	X	X	O	X	X	O	X	X	X	X	X	X	X	X	X	X	X	X	X	X	X	X	X	X	X	X	X	X	X	X	X	X	X	X	X	X	X	X	X	X	X	X	0.38
	$R_{2}^{Sf}$	O	X	X	O	X	X	O	X	X	O	X	X	O	X	X	O	X	X	O	X	X	O	X	X	O	X	X	O	X	X	O	X	X	O	X	X	X	X	X	X	X	X	X	X	X	X	0.38
	$R_{3}^{Sf}$	O	X	X	O	X	X	O	X	X	O	X	X	O	X	X	O	X	X	O	X	X	O	X	X	O	X	X	O	X	X	O	X	X	O	X	X	O	X	X	O	X	X	O	X	X	O	0.38

**Table 21 sensors-24-03881-t021:** Probabilistic requirements verification table (improved to AmbX_4_).

Requirements		Va Probabilities	System Probabilities	Verification Results	
$R^{Sc}$	$R_{1}^{Sc}$	0.35	0.38	O	Satisfied
	$R_{2}^{Sc}$	0.40	0.38	X	Unsatisfied, but Satisfiable
	$R_{3}^{Sc}$	0.30	0.32	O	Satisfied
$R^{Sf}$	$R_{1}^{Sf}$	0.50	0.38	X	Unsatisfied, but Satisfiable
	$R_{2}^{Sf}$	0.30	0.38	O	Satisfied
	$R_{3}^{Sf}$	0.20	0.38	O	Satisfied

**Table 22 sensors-24-03881-t022:** Probabilistic equivalences in the smart EMS example.

Process	Probabilistic Process Equivalence	Imterpertation
HospM	${HospM}_{1}^{\langle 0.9\rangle }\overset{\Delta^{\langle 0.05\rangle }}{\sim }{HospM}_{3}^{\langle 0.95\rangle }$	0.05% improvement from HospM_1_ to HospM_3_
AmbX	${AmbX}_{1}^{\langle 0.9,0.9\rangle }\overset{\Delta^{\langle 0.05,0.05\rangle }}{\sim }{AmbX}_{2}^{\langle 0.95,0.95\rangle }$	0.05% improvement from 911Center_1_ to 911Center_3_
911Center	${911Center}_{1}^{\langle 0.9,0.7\rangle }\overset{\Delta^{\langle 0.05,0.24\rangle }}{\sim }{911Center}_{3}^{\langle 0.95,0.94\rangle }$	0.05% improvement from AmbX_1_ to AmbX_2_
AmbY	${AmbY}_{1}^{\langle 0.7,0.9\rangle }\overset{\Delta^{\langle 0.25,0.05\rangle }}{\sim }{AmbY}_{3}^{\langle 0.95,0.95\rangle }$	0.25% improvement from AmbY_1_ to AmbY_3_
HospN	${HospN}_{1}^{\langle 0.9\rangle }\overset{\Delta^{\langle 0.05\rangle }}{\sim }{HospN}_{2}^{\langle 0.95\rangle }$	0.05% improvement from HospN_1_ to HospN_2_

**Table 23 sensors-24-03881-t023:** Probabilistic reachability table for the smart EMS system S_1_ (HospM_1_, AmbX_1_, 911Center_1_, AmbY_1_, HospN_1_) (Same as Table 18).

BR	Reachability Table																																													
1	0.9																							0.1																						
2	0.9																						0.1	0.9	0.1																					
3	0.9											0.1													0.9											0.1										
4	0.9										0.1	0.9	0.1												0.9										0.1	0.9	0.1									
5	0.7					0.3							0.7					0.3							0.7					0.3							0.7					0.3				
6	0.7				0.3	0.7	0.3						0.7				0.3	0.7	0.3						0.7				0.3	0.7	0.3						0.7				0.3	0.7	0.3			
7	0.9		0.1				0.9		0.1				0.9		0.1				0.9		0.1				0.9		0.1				0.9		0.1				0.9		0.1				0.9		0.1	
8	0.9	0.1	0.9	0.1			0.9	0.1	0.9	0.1			0.9	0.1	0.9	0.1			0.9	0.1	0.9	0.1			0.9	0.1	0.9	0.1			0.9	0.1	0.9	0.1			0.9	0.1	0.9	0.1			0.9	0.1	0.9	0.1
Path	1	2	3	4	5	6	7	8	9	10	11	12	13	14	15	16	17	18	19	20	21	22	23	24	25	26	27	28	29	30	31	32	33	34	35	36	37	38	39	40	41	42	43	44	45	46
Prob.	0.26	0.03	0.03	0.00	0.14	0.14	0.06	0.01	0.01	0.00	0.07	0.07	0.00	0.00	0.00	0.00	0.00	0.00	0.00	0.00	0.00	0.00	0.09	0.09	0.00	0.00	0.00	0.00	0.00	0.00	0.00	0.00	0.00	0.00	0.00	0.00	0.00	0.00	0.00	0.00	0.00	0.00	0.00	0.00	0.00	0.00
Total	0.32 (Success) + 0.68 (Fail) = 1																																													

The background color in this table indicates deadlock cases.

**Table 24 sensors-24-03881-t024:** Probabilistic reachability table for the smart EMS system S_3_ (HospM_3_, AmbX_2_, 911Center_3_, AmbY_3_, HospN_2_).

BR	Reachability Table																																													
1	0.95																							0.05																						
2	0.95																						0.05	0.95	0.05																					
3	0.95											0.05													0.95											0.05										
4	0.95										0.05	0.95	0.05												0.95										0.05	0.95	0.05									
5	0.94					0.06							0.94					0.06							0.94					0.06							0.94					0.06				
6	0.95				0.05	0.95	0.05						0.95				0.05	0.95	0.05						0.95				0.05	0.95	0.05						0.95				0.05	0.95	0.05			
7	0.95		0.05				0.95		0.05				0.95		0.05				0.95		0.05				0.95		0.05				0.95		0.05				0.95		0.05				0.95		0.05	
8	0.9	0.1	0.9	0.1			0.9	0.1	0.9	0.1			0.9	0.1	0.9	0.1			0.9	0.1	0.9	0.1			0.9	0.1	0.9	0.1			0.9	0.1	0.9	0.1			0.9	0.1	0.9	0.1			0.9	0.1	0.9	0.1
Path	1	2	3	4	5	6	7	8	9	10	11	12	13	14	15	16	17	18	19	20	21	22	23	24	25	26	27	28	29	30	31	32	33	34	35	36	37	38	39	40	41	42	43	44	45	46
Prob.	0.64	0.04	0.04	0.01	0.04	0.04	0.01	0.00	0.00	0.00	0.04	0.04	0.00	0.00	0.00	0.00	0.00	0.00	0.00	0.00	0.00	0.00	0.05	0.05	0.00	0.00	0.00	0.00	0.00	0.00	0.00	0.00	0.00	0.00	0.00	0.00	0.00	0.00	0.00	0.00	0.00	0.00	0.00	0.00	0.00	0.00
Total	0.66 (Success) + 0.34 (Fail) = 1																																													

The background color in this table indicates deadlock cases.

**Table 25 sensors-24-03881-t025:** Analysis of the system probability for the smart EMS example (multiple improvements).

Path		1	2	3	4	5	6	7	8	9	10	11	12	13	14	15	16	17	18	19	20	21	22	23	24	25	26	27	28	29	30	31	32	33	34	35	36	37	38	39	40	41	42	43	44	45	46	Prob.
Prob.	Succ.	0.64			0.01			0.01			0.00			0.00			0.00			0.00			0.00			0.00			0.00			0.00			0.00			0.00			0.00			0.00			0.00	0.66
	Fail		0.04	0.04		0.04	0.04		0.00	0.00		0.04	0.04		0.00	0.00		0.00	0.00		0.00	0.00		0.05	0.05		0.00	0.00		0.00	0.00		0.00	0.00		0.00	0.00		0.00	0.00		0.00	0.00		0.00	0.00		0.34
$R^{Sc}$	$R_{1}^{Sc}$	O	X	X	O	X	X	O	X	X	O	X	X	O	X	X	O	X	X	O	X	X	O	X	X	O	X	X	O	X	X	O	X	X	O	X	X	O	X	X	O	X	X	O	X	X	O	0.66
	$R_{2}^{Sc}$	O	X	X	O	X	X	O	X	X	O	X	X	O	X	X	O	X	X	O	X	X	O	X	X	X	X	X	X	X	X	X	X	X	X	X	X	X	X	X	X	X	X	X	X	X	X	0.66
	$R_{3}^{Sc}$	O	X	X	O	X	X	X	X	X	X	X	X	O	X	X	O	X	X	X	X	X	X	X	X	O	X	X	O	X	X	X	X	X	X	X	X	O	X	X	O	X	X	X	X	X	X	0.66
$R^{Sf}$	$R_{1}^{Sf}$	O	X	X	O	X	X	O	X	X	O	X	X	X	X	X	X	X	X	X	X	X	X	X	X	X	X	X	X	X	X	X	X	X	X	X	X	X	X	X	X	X	X	X	X	X	X	0.65
	$R_{2}^{Sf}$	O	X	X	O	X	X	O	X	X	O	X	X	O	X	X	O	X	X	O	X	X	O	X	X	O	X	X	O	X	X	O	X	X	O	X	X	X	X	X	X	X	X	X	X	X	X	0.66
	$R_{3}^{Sf}$	O	X	X	O	X	X	O	X	X	O	X	X	O	X	X	O	X	X	O	X	X	O	X	X	O	X	X	O	X	X	O	X	X	O	X	X	O	X	X	O	X	X	O	X	X	O	0.66

**Table 26 sensors-24-03881-t026:** Validation results for the probabilistic requirements (multiple improvements).

Requirements		Va Probabilities	System Probabilities	Verification Results	
$R^{Sc}$	$R_{1}^{Sc}$	0.35	0.66	O	Satisfied
	$R_{2}^{Sc}$	0.40	0.66	O	Satisfied
	$R_{3}^{Sc}$	0.30	0.66	O	Satisfied
$R^{Sf}$	$R_{1}^{Sf}$	0.50	0.65	O	Satisfied
	$R_{2}^{Sf}$	0.30	0.66	O	Satisfied
	$R_{3}^{Sf}$	0.20	0.66	O	Satisfied

**Table 27 sensors-24-03881-t027:** Differences between previous research and the present research.

	Previous Research	Present Research
Probability Enhancement	Arbitrary	Deterministic
Incremental Improvement	Arbitrary	Step-wise

**Table 28 sensors-24-03881-t028:** Differences between related research and the present research.

	Other Research	Present Research
Probabilistic Equivalence	System-Level Probabilistic Equivalence Continuous and Sequential Equivalence	Process Equivalence: Incremental and Discrete Equivalence System Equivalence: Step-wise Integrated Equivalence

## Data Availability

Data are contained within the article.

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
