# Peer review of "A Process Algebraic Approach to Predict and Control Uncertainty in Smart IoT Systems for Smart Cities Based on Permissible Probabilistic Equivalence"

_sensors, 2024, doi:10.3390/s24123881_

Round 1

Reviewer 1 Report

Comments and Suggestions for Authors

The work looks somewhat divorced from reality.
The very idea of using visual notation requires a fairly deep justification.
However, the development of the dTP-Calculus formalism looks quite interesting.

Author Response

Submitting the response to the paper sensors-3005376.
Please see the attachment. 

Reviewer 2 Report

Comments and Suggestions for Authors

1) All acronyms must be written in full the first time they appear within text.

2) You need to revise the abstract so that it is concise, flowing and clear to the readers.

3) Towards the end of the abstract, provide a summary of the study findings.

4) In as much as possible, avoid listing of the key points/concepts and instead, discuss them in continuous paragraphs

5) In sub-section 1.6, you write as follows:

"....The methodology and approach can be considered to be one of the most efficient practices in the area of process algebra to predict and control uncertainty and risks caused by nondeterministic behavior of Smart IoT Systems with probability...."

You need to support the 'efficiency' aspect of the above claim by some data.

6) This paper is unnecessarily long. You need to shorten it by presenting and discussing only the novel aspects. Well known concepts need not be discussed in detail. For instance, you have included many screenshots which can be replaced by concise flow diagrams/pseudocodes.

7) The clarity of all figures need to be improved.

8) The entire paper need to be reorganized. As it stands, there seem to be a mixture of results and methodology in some sections.

9) You seem to base most of your arguments in your previous work. Aren't there other related works in this domain from which comparative evaluations can be drawn?

10)  The conclusion section is not well done. Revise it in such a way that it is in line with the study objectives and the obtained data.

Comments on the Quality of English Language

English used throughout the manuscript is fair.

Author Response

(The authors gave the same response as above.)

Reviewer 3 Report

Comments and Suggestions for Authors

A Process Algebraic Approach to Predict and Control Uncertainty in Smart IoT Systems for Smart Cities Based on Permissible Probabilistic Equivalence

Recommendation for the editor

Accept after revision 

Review

This paper analyzes the controllability of nondeterministic behavior in IoT systems using the dTP-Calculus developed in previous work. The dTP-Calculus addresses this aspect by verifying static and dynamic probabilities of security requirements using discrete and incremental probabilities to control nondeterminism and satisfy requirements through process equivalence methods. The SAVE tool in the ADOxx Meta-Modeling Platform demonstrates the approach's feasibility and practicality in analyzing an example EMS Smart System, showing that it can lead to good practice for predicting and controlling uncertainty in IoT Smart Systems.

 However, there are still some questions that remain.

1.    Beyond the properties of dTP-Calculus of mobility, temporality, and probability [section 2.1.1], what other quantitative system properties, like reliability, robustness, and performance, can IoT modeling obtain through the dTP-Calculus?

2.    In section 2.2, “Probabilistic Equivalences” [line 457], the tested processes are structurally equivalent, differing only in the probabilities of choice as required for the difference between probabilities [lines 483-492]. However, structural equivalence is not required for other process equivalences (like bisimulation or testing equivalence). This raises interesting questions: what might it mean to compare, for example, strongly bisimilar processes for probabilistic equivalence when they are not structurally equivalent? How does this affect the proposed probabilistic equivalence procedure? Could other forms of probabilistic equivalence help in System Improvement as defined in section 3.3 [lines 602-604]?

3.    For Process Enhancement [lines 600-602], how can efficiently be achieved the proper adjustment in the values of the probabilities in transitions? No automated procedure is described for this purpose, as declared in section 3.3 [lines 591-599].

4.    Does Probabilistic Equivalence provide compositional and incremental design and verification? As systems grow in complexity, being able to verify components or aspects of the system separately and then compose these verified parts into a whole while maintaining correctness could significantly improve development efforts. Is it possible to change a process by another that is bisimilar but not structurally equivalent?

5.    Following the reasoning posed by question 3, can the SAVE tool [section 4.2.3, lines 710-726] be affordably enhanced to fully provide automated synthesis and control under uncertainty for Process Enhancement and Improvement? The problem of automatically synthesizing controllers for timed probabilistic systems, especially in uncertainty and partial observability, is still open.

6.    How does this work compare to others (like PRISM, MRMC, etc.) through benchmarking? How can we choose the best approach for probabilistic model checking if benchmarks and metrics for evaluating and comparing methods and tools are missing? 

7.    How expressive is the dTP-Calculus language for suitable formal description in specific areas like IoT? Developing methodologies for specific application domains, such as IoT, presents significant challenges. The IoT domain has unique requirements and constraints that general-purpose methods may not effectively address. Several concerns like unreliable communication (frequent disconnections, fast change in network topology, etc.), sudden shutdown due to battery exhaustion, and unpredictable arrival times in heavy traffic conditions, among others, make it challenging to predict system behavior and performance. As a concrete example for the EMS case study, a frequent concern during the COVID pandemic was changing the selected hospital for an emergency due to insufficient hospital resources or excessive traffic conditions, and still, trying to ensure that ambulances arrived on time. How can this work help in this kind of problem?

Author Response

(The authors gave the same response as above.)
